# Optimizing importance weighting in the presence of sub-population shifts

**Floris Holstege**[1,2]**, Bram Wouters**[1]**, Noud van Giersbergen**[1]**, Cees Diks**[1,2]

[1]University of Amsterdam, Department of Quantitative Economics    [2]Tinbergen Institute

`{f.g.holstege,b.m.wouters,n.p.a.vangiersbergen,c.g.h.diks}@uva.nl`

## Abstract

A distribution shift between the training and test data can severely harm performance of machine learning models. Importance weighting addresses this issue by assigning different weights to data points during training. We argue that existing heuristics for determining the weights are suboptimal, as they neglect the increase of the variance of the estimated model due to the finite sample size of the training data. We interpret the optimal weights in terms of a bias-variance trade-off, and propose a bi-level optimization procedure in which the weights and model parameters are optimized simultaneously. We apply this optimization to existing importance weighting techniques for last-layer retraining of deep neural networks in the presence of sub-population shifts and show empirically that optimizing weights significantly improves generalization performance.

## 1 Introduction

Machine learning (ML) models typically are trained to minimize the average loss on training data, with the implicit aim to generalize well to unseen test data. A distribution shift between the training and test data can severely harm generalization performance (Alcorn et al., 2018; Quiñonero-Candela et al., 2022). Distribution shifts come in many forms; a prime example is a so-called sub-population shift, where the population is dissected in groups and the distribution shift is entirely due to a change in occurrence frequencies of the different groups (Yang et al., 2023). In ML applications, e.g., in the medical field (Jabbour et al., 2020), such shifts can negatively impact the average accuracy on test data and/or the accuracy on specific groups. The latter has implications for algorithmic fairness (Shankar et al., 2017; Buolamwini & Gebru, 2018; Wilson et al., 2019). Furthermore, sub-population shifts are also relevant regarding spurious correlations (Gururangan et al., 2018; Geirhos et al., 2020; DeGrave et al., 2021), where the frequency of occurrence of certain combinations of the variable of interest (e.g., a medical condition) and non-causally related attributes (e.g., the sex of the patient) might differ between the training and test distribution.

Importance weighting applied to ML models treats different observations in the training data differently, for example, by using a weighted loss or a weighted sampler (Kahn & Marshall, 1953). It has long been used to address distribution shifts and other issues, including domain adaptation, covariate shift and class imbalance; see Kimura & Hino (2024) for an overview. In the context of training deep neural networks (DNNs), it was used in a variety of ways to tackle sub-population shifts, e.g., Sagawa et al. (2020a); Liu et al. (2021); Idrissi et al. (2022); Kirichenko et al. (2023).

This paper addresses the question of how to weigh each observation in an importance weighting scheme that aims to mitigate the effects of a distribution shift. Typically, the weights are chosen based on a heuristic argument. For example, in group-weighted empirical risk minimization (GW-ERM) they are chosen such that the weighted training loss is unbiased for the expected loss with respect to the test distribution. We argue that such a choice is usually not optimal: for a limited training sample it does not lead to estimated model parameters that minimize the expected loss with respect to the test distribution. Because of the limited size of the sample, heuristically chosen weights tend to put too much weight on certain observations (usually the minority groups). They thereby introduce significant variance, due to a strong dependence of the trained model on the particulars of the training data. We illustrate this bias-variance trade-off for importance weighting analytically with an example of a linear regression model in the presence of a sub-population shift.

Secondly, we introduce an estimation procedure for the optimal weights via a bi-level optimization problem, where the weights and the model parameters are optimized simultaneously. In the context of last-layer retraining of DNNs in the presence of sub-population shifts, we show empirically, for benchmark vision and natural language processing datasets, that existing state-of-the-art importance weighting methods (see Section 2) can be improved significantly by optimizing for the weights. We find that both the weighted average and worst-group accuracy generally improve, and that optimized weights increase robustness against the choice of hyperparameters for training. We show that the effect of optimizing weights is larger when the limited size of the training sample becomes pressing, namely in the case of a small total sample size or when the minority groups are small. We provide an implementation of this procedure as an open-source package (link).

We emphasize that the proposed procedure for optimizing weights should not be positioned as a new importance weighting method. Instead, it should be seen as an addition to existing methods that typically improves their generalization performance.

## 2 RELATED WORK

In this section we give a brief overview of importance weighting techniques, the issue of sub-population shift, and how existing methods aim to address it.

**Importance weighting for distribution shifts in deep learning:** importance weighting is a classical tool to estimate a model for a test distribution that differs from the distribution of the training sample (Kahn & Marshall, 1953; Shimodaira, 2000; Koller & Friedman, 2009). When applied to training DNNs, importance weighting only has a noticeable effect in the presence of sufficient regularization (Byrd & Lipton, 2018; Xu et al., 2021). In our applications we focus on the use of importance weighting for last-layer retraining, as this has been argued to be sufficient for dealing with sub-population shifts (Kirichenko et al., 2023; LaBonte et al., 2023; Chen et al., 2024).

**Sub-population shifts:** it has been widely documented that ML models frequently face deteriorating generalization performance due to sub-population shifts (Cai et al., 2021; Koh et al., 2021; Yang et al., 2023). There exist different possible definitions of the groups for which the distribution shift occurs. In the case of a spurious correlation, groups are commonly defined by a combination of a label and a non-causally related attribute (Joshi et al., 2022; Makar et al., 2022). Alternatively, groups can be defined exclusively via an attribute (Martinez et al., 2021b) or, in the case of class imbalance, exclusively via a label (Liu et al., 2019). While our experiments consider spurious correlations, the methodology also applies to alternative group definitions. Generalizing to groups that do not occur in the training data (Santurkar et al., 2020) is beyond the scope of this paper.

**Existing approaches of importance weighting for training DNNs in the presence of a sub-population shift:** a key method is group-weighted empirical risk minimization (GW-ERM), which uses a weighted loss and where the weights are determined by the occurrence frequencies of the groups in the training and test distribution. The weighted sampler analogue subsamples from large groups (SUBG) to create a novel dataset (Sagawa et al., 2020b). Both methods are strong baselines for addressing sub-population shifts (Idrissi et al., 2022). An alternative approach is to minimize the worst-group loss via group distributional robust optimization (GDRO, Sagawa et al. (2020a)). While these methods require group annotations for all training data, there exist other methods that require limited group annotations in the form of a validation set. An example is Just Train Twice (JTT), where the errors of the model are used to identify data points that need to be upweighted (Liu et al., 2021). Other methods in this realm incorporate errors of a secondary model in their estimation procedure (Nam et al., 2020; Zhang et al., 2022; Qiu et al., 2023) or estimate group attributes (e.g., Nam et al. (2022); Han & Zou (2024); Pezeshki et al. (2024)). These methods typically require a small validation set for determining hyperparameters. Kirichenko et al. (2023) argue that instead one should use this validation set for the estimation of an ensemble of models via SUBG, a method denoted as deep feature reweighting (DFR).

**Estimation of optimal weights:** a commonly noted downside of importance weighting is that it reduces the effective sample size available for model training (Martino et al., 2017; Schwartz & Stanovsky, 2022). In light of this, Cui et al. (2019) and Wen et al. (2020) suggest that the weight given to a particular label should be based on the number of available datapoints. Similar to our approach, Ren et al. (2018) aim to learn weights for neural networks given a validation set.

## 3 PROBLEM SETTING

Consider random variables $(y, \boldsymbol{x})$, where $y \in \mathcal{Y}$ is the target variable and $\boldsymbol{x} \in \mathcal{X}$ represents the input features. Suppose we are dealing with a family of models $\{f_{\boldsymbol{\theta}} : \mathcal{X} \to \mathcal{Y} \,|\, \boldsymbol{\theta} \in \Theta\}$ and we aim to find a model that minimizes $\mathbb{E}_{(y,\boldsymbol{x}) \sim \mathcal{P}_{\mathrm{te}}}[\mathcal{L}(y, f_{\boldsymbol{\theta}}(\boldsymbol{x}))]$, where $\mathcal{L}(\cdot, \cdot)$ is some pre-defined loss function and $\mathcal{P}_{\mathrm{te}}$ is a test distribution of interest. Moreover, we have access to a dataset $D_{\mathrm{tr}}^n = \{y_i, \boldsymbol{x}_i\}_{i=1}^n$, which is an i.i.d. sample drawn from a training distribution $\mathcal{P}_{\mathrm{tr}}$ that is different from the test distribution. This distribution shift can be problematic, as the minimum of an empirical loss for the training data does typically not coincide with the minimum expected loss with respect to the test distribution.

One method to address this issue is by estimating the model parameters using a weighted loss,

$$\hat{\boldsymbol{\theta}}_n(\boldsymbol{w}) := \arg\min_{\boldsymbol{\theta} \in \Theta} \mathcal{L}_{\mathrm{train}}^n(\boldsymbol{\theta}, \boldsymbol{w}), \qquad \text{where} \quad \mathcal{L}_{\mathrm{train}}^n(\boldsymbol{\theta}, \boldsymbol{w}) := \frac{1}{n} \sum_{i=1}^n w_i \, \mathcal{L}(y_i, f_{\boldsymbol{\theta}}(\boldsymbol{x}_i)). \tag{1}$$

If there is no unique minimizer, which is often the case for DNNs, the $\arg\min$ should be interpreted as drawing randomly one element from the set of minimizers. A common choice for the weights $\boldsymbol{w} = (w_1, w_2, \ldots w_n)' \in \mathbb{R}^n$ are the likelihood ratios $\boldsymbol{r} = (r_1, r_2, \ldots r_n)'$, where $r_i = r(y_i, \boldsymbol{x}_i)$ and $r(y, \boldsymbol{x}) := \frac{p_{\mathrm{te}}(y, \boldsymbol{x})}{p_{\mathrm{tr}}(y, \boldsymbol{x})}$ is the (assumed to be known) ratio of the probability density functions (pdfs) associated with the test and training distribution respectively. This is due to a key result in importance weighting that, assuming $p_{\mathrm{tr}}(y, \boldsymbol{x}) > 0$ when $p_{\mathrm{te}}(y, \boldsymbol{x}) \neq 0$,

$$\mathbb{E}_{(y,\boldsymbol{x}) \sim \mathcal{P}_{\mathrm{tr}}}[r(y, \boldsymbol{x})\mathcal{L}(y, f_{\boldsymbol{\theta}}(\boldsymbol{x}))] = \mathbb{E}_{(y,\boldsymbol{x}) \sim \mathcal{P}_{\mathrm{te}}}[\mathcal{L}(y, f_{\boldsymbol{\theta}}(\boldsymbol{x}))]. \tag{2}$$

If we choose the weights $\boldsymbol{w}$ to be equal to the likelihood ratios $\boldsymbol{r}$, then by the law of large numbers $\mathcal{L}_{\mathrm{train}}^n(\boldsymbol{\theta}, \boldsymbol{r})$ converges in the limit of $n \to \infty$ to the expectations in Equation 2. As a consequence, $\hat{\boldsymbol{\theta}}_n(\boldsymbol{r})$ minimizes for sample size tending to infinity the expected loss with respect to the test distribution, $\mathbb{E}_{(y,\boldsymbol{x}) \sim \mathcal{P}_{\mathrm{te}}}[\mathcal{L}(y, f_{\boldsymbol{\theta}}(\boldsymbol{x})]$. However, in practice one works with a finite amount of training data and the expectation might be poorly approximated by its sample analogue. Therefore, the optimal choice $\boldsymbol{w}_n^*$ for the weights should be defined as the one that leads to a "best" parameter estimate $\hat{\boldsymbol{\theta}}_n(\boldsymbol{w})$, in the sense that it minimizes the expected loss with respect to the test distribution,

$$\boldsymbol{w}_n^* := \arg\min_{\boldsymbol{w}} \mathbb{E}_{D_{\mathrm{tr}}^n \sim \mathcal{P}_{\mathrm{tr}}} \left[ \mathbb{E}_{(y,\boldsymbol{x}) \sim \mathcal{P}_{\mathrm{te}}}[\mathcal{L}(y, f_{\hat{\boldsymbol{\theta}}_n(\boldsymbol{w})}(\boldsymbol{x}))|D_{\mathrm{tr}}^n] \right]. \tag{3}$$

Note that by marginalizing over $D_{\mathrm{tr}}^n$ the optimal choice for $\boldsymbol{w}$ is not determined by the actual realization of the training data. In practice, with access to only a single dataset $D_{\mathrm{tr}}^n$, the estimate of the optimal choice will be optimal given that specific training set, although one might be able to alleviate this dependence by using a cross-validation or bootstrapping scheme.

The first main claim of this paper is that the optimal weights $\boldsymbol{w}_n^*$ are typically not equal to the weight choice based on the likelihood ratio, $(\boldsymbol{w}_n^*)_i \neq r(y_i, \boldsymbol{x}_i)$. Although the likelihood ratio makes the estimated parameters asymptotically unbiased under certain conditions (Shimodaira (2000)), it neglects the variance that arises due to a limited size of the training set. More concretely, putting more weight on certain training data points could lead to overfitting and does not necessarily lead to better generalization, not even on the test distribution for which the weights are supposed to be a correction.

The second main contribution of this paper is the introduction of a new estimation procedure for the optimal weights $\boldsymbol{w}_n^*$. To address the expectation with respect to $\mathcal{P}_{\mathrm{te}}$ in Equation 3, we make use of Equation 2 and write the optimal weights as

$$\boldsymbol{w}_n^* = \arg\min_{\boldsymbol{w}} \mathbb{E}_{D_{\mathrm{tr}}^n \sim \mathcal{P}_{\mathrm{tr}}} \left[ \mathbb{E}_{(y,\boldsymbol{x}) \sim \mathcal{P}_{\mathrm{tr}}}[r(y, \boldsymbol{x})\mathcal{L}(y, f_{\hat{\boldsymbol{\theta}}_n(\boldsymbol{w})}(\boldsymbol{x}))|D_{\mathrm{tr}}^n] \right]. \tag{4}$$

We propose to estimate the inner expectation using a randomly drawn subset of size $n_{\mathrm{val}}$ from the available training data. We call this subset the validation set. Let $n' = n - n_{\mathrm{val}}$ be the remaining number of data points. The result is a bi-level optimization problem. The weights are estimated via

$$\hat{\boldsymbol{w}}_n^* = \arg\min_{\boldsymbol{w}} \mathcal{L}_{\mathrm{val}}(\hat{\boldsymbol{\theta}}_{n'}(\boldsymbol{w}), \boldsymbol{r}), \tag{5}$$

where the validation loss is defined analogously to the training loss in Equation 1. The model parameters are estimated by minimizing this training loss, using the $n'$ data points that are not in the validation set. The creation of the validation set has the advantage of avoiding possible overfitting of $\boldsymbol{w}$, but comes at the cost of having fewer data points for estimating $\boldsymbol{\theta}$.

### 3.1 OPTIMIZING WEIGHTS FOR SUB-POPULATION SHIFTS

In what follows, we make the previous general claims concrete by focusing on one specific type of distribution shift that has been studied extensively in the machine learning literature, namely sub-population shift. Each observation carries an additional random variable $g \in \mathcal{G}$, which labels a total number of $|\mathcal{G}| = G$ subgroups that dissect the population. We note that $g$ can depend on $y$ and $\boldsymbol{x}$, but this is not necessarily the case. For ease of notation, we assume that $g$ can take the values $1, 2, \ldots, G$. The distribution shift arises from a difference between the marginal training and test distributions of $g$, while the conditional pdf $p(\boldsymbol{x}, y|g)$ remains unchanged:

$$p_{\text{tr}}(y, g, \boldsymbol{x}) = p_{\text{tr}}(g)p(y, \boldsymbol{x}|g), \quad p_{\text{te}}(y, g, \boldsymbol{x}) = p_{\text{te}}(g)p(y, \boldsymbol{x}|g), \tag{6}$$

where $p_{\text{tr}}(g) \neq p_{\text{te}}(g)$. Unless otherwise stated, for the training set defined in Section 3 we presume access to the subgroup labels $\{g_i\}_{i=1}^n$.

Since the distribution shift is entirely due to a change in the marginal distribution of $g$, it is natural to constrain the weights $w_i$ to be determined by the group labels $g_i$. It is convenient to define group weights $p_g$, where $g \in \mathcal{G}$, such that $w_i = p_{g_i}/p_{\text{tr}}(g_i)$. Estimating the model parameters can then be seen as a function of the group weights, i.e., $\hat{\boldsymbol{\theta}}_n(\boldsymbol{p})$, where $\boldsymbol{p} = (p_1, p_2, \ldots, p_G)' \in \mathbb{R}_+^G$. Instead of optimal weights $\boldsymbol{w}_n^*$, one now aims to find optimal group weights $\boldsymbol{p}_n^*$. Assuming that the number of groups is much smaller than the size of the training set, this heavily reduces the dimension of the bi-level optimization problem defined in Equation 5.

## 4 THEORETICAL ANALYSIS OF LINEAR REGRESSION

The optimal weights, as defined in Equation 3, can be analytically derived for the case of linear regression with a simple data-generating process (DGP) and sub-population shift. This analysis provides insight into the nature of optimized weights as a bias-variance trade-off and the conditions under which optimization of the weights becomes relevant.

We consider a DGP $y = \boldsymbol{x}^{\text{T}}\boldsymbol{\beta}^g + \varepsilon$, where $y \in \mathbb{R}$ is the target variable, $\boldsymbol{x} = (1, \tilde{\boldsymbol{x}}^{\text{T}})^{\text{T}}$ represents the intercept and the feature vector $\tilde{\boldsymbol{x}} \in \mathbb{R}^d$, and $\varepsilon \in \mathbb{R}$ is the noise. We take $\tilde{\boldsymbol{x}} \sim \mathcal{N}(\boldsymbol{0}, \boldsymbol{I}_d)$ and the noise to have zero mean, variance $\sigma^2$ and to be uncorrelated with $\boldsymbol{x}$. The population consists of two groups labeled by $g \in \{0, 1\}$. Group membership determines the value of the intercept, $\boldsymbol{\beta}^g = (a_g, \boldsymbol{\beta}^{\text{T}})^{\text{T}}$ for $g = 0, 1$. Note that the slope coefficients are common among the two groups. The sub-population shift between training and test data is defined by a change in the marginal distribution of the group label, $p_{\text{tr}} := \mathbb{P}_{\text{tr}}(g = 1) \neq \mathbb{P}_{\text{te}}(g = 1) =: p_{\text{te}}$.

Suppose we have a training sample $D_{\text{tr}}^n = \{y_i, \boldsymbol{x}_i, g_i\}_{i=1}^n$ of size $n$ and we use the weighted least squares (WLS) estimator

$$\hat{\boldsymbol{\beta}}_n(p) = (\boldsymbol{X}^{\text{T}}\boldsymbol{W}\boldsymbol{X})^{-1}\boldsymbol{X}^{\text{T}}\boldsymbol{W}\boldsymbol{y}, \quad \text{with } \boldsymbol{W} = \text{diag}(w_1, ..., w_n), \ w_i = \begin{cases} \frac{p}{p_{\text{tr}}} & g_i = 1, \\ \frac{1-p}{1-p_{\text{tr}}} & g_i = 0. \end{cases} \tag{7}$$

Here, $\boldsymbol{X} \in \mathbb{R}^{n \times (d+1)}$ is the design matrix of input features and $\boldsymbol{y} \in \mathbb{R}^n$ contains the target variables. The weights $w_i$ are parameterized by $p \in [0, 1]$. If we choose the weights to be the likelihood ratio between the test and training distribution, the WLS estimator asymptotically minimizes the expected quadratic loss with respect to the test distribution. This choice of the weights corresponds to $p = p_{\text{te}}$.

For large but finite $n$, the approximate expected loss is given by (see Appendix B for details)

$$\mathbb{E}_{D_{\text{tr}}^n \sim \mathcal{P}_{\text{tr}}}\left[\mathbb{E}_{(y, \boldsymbol{x}) \sim \mathcal{P}_{\text{te}}}\left[(y - \boldsymbol{x}^{\text{T}}\hat{\boldsymbol{\beta}}_n(\boldsymbol{p}))^2\middle|D_{\text{tr}}^n\right]\right] \approx \mathcal{B}^2(p_{\text{te}}, p, a_1, a_0) + \mathcal{V}(\sigma^2, p, p_{\text{tr}}, n, d) + \sigma^2, \tag{8a}$$

where the result is decomposed in the following bias and variance term, respectively,

$$\mathcal{B}^2(p_{\text{te}}, p, a_1, a_0) = \left[p_{\text{te}}(1-p)^2 + (1-p_{\text{te}})p^2\right](a_1 - a_0)^2, \tag{8b}$$

$$\mathcal{V}(\sigma^2, p, p_{\text{tr}}, n, d) = \sigma^2\left[\frac{p^2}{p_{\text{tr}}} + \frac{(1-p)^2}{1-p_{\text{tr}}}\right]\frac{d+1}{n}. \tag{8c}$$

Both terms are quadratic with respect to $p$. The bias term is minimized for $p = p_{\text{te}}$, provided that $a_0 \neq a_1$, whereas the variance is minimized for $p = p_{\text{tr}}$. The minimum of the expected loss is at

$$p_n^* = \frac{p_{\text{te}} + \eta\, p_{\text{tr}}}{1 + \eta}, \quad \text{where } \eta = \frac{\sigma^2(d+1)}{n(a_1 - a_0)^2 p_{\text{tr}}(1 - p_{\text{tr}})}. \tag{9}$$

This is the optimal trade-off between minimizing the bias and variance. In the limit $n \to \infty$ the variance disappears and the optimal $p$ is indeed equal to the heuristic choice $p_{\text{te}}$. However, for finite $n$ it can differ considerably. This is illustrated in Figure 1. Its impact on the expected loss is shown in Figure 5 in Appendix B.

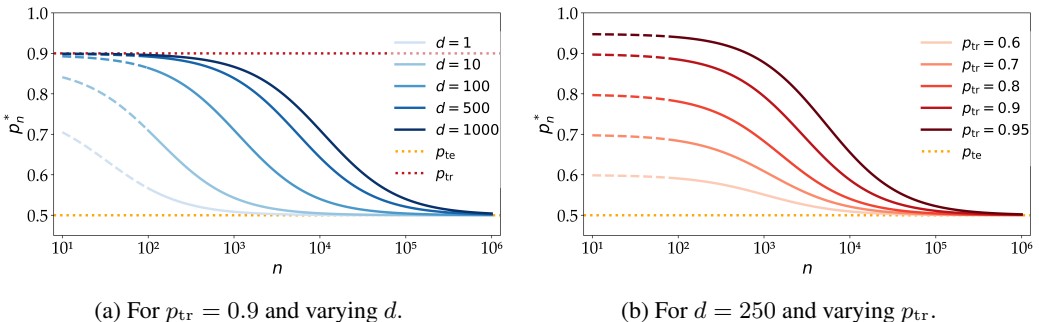

(a) For $p_{\text{tr}} = 0.9$ and varying $d$.      (b) For $d = 250$ and varying $p_{\text{tr}}$.

Figure 1: The optimal choice $p_n^*$, given by Equation 9, as a function of the training dataset size $n$, for varying feature dimension $d$ (panel (a)) and training distribution $p_{\text{tr}}$ (panel (b)). Other parameters are fixed at $a_1 = 1, a_0 = 0, \sigma^2 = 1, p_{\text{te}} = 0.5$. Note that the heuristic choice for $p$ would be at 0.5. Also note that Equation 9 is derived approximately for large $n$ (hence the dashed lines for small $n$).

## 5 OPTIMIZING WEIGHTS OF EXISTING IMPORTANCE WEIGHTING TECHNIQUES FOR SUB-POPULATION SHIFTS

Before continuing, we note that the setup described in Section 3, when applied to sub-population shifts, largely corresponds to GW-ERM. However, as emphasized in Section 1, the methodology proposed in this paper also applies to other importance weighting methods. They differ from GW-ERM in terms of, e.g., the use of a weighted sampler instead of a weighted loss (SUBG, DFR), the overall objective (GDRO) or the inference of groups (JTT). For all these cases, optimal weights can be defined and subsequently found through a bi-level optimization problem. In what follows, we discuss the details of optimizing weights for the different existing importance weighting techniques.

### 5.1 GROUP-WEIGHTED EMPIRICAL RISK MINIMIZATION

The optimal group weights $\boldsymbol{p}_n^*$ are estimated by solving a bi-level optimization problem that for GW-ERM consists of parameter estimation as in Equation 1, with the difference that the estimates are a function of $\boldsymbol{p}$ instead of $\boldsymbol{w}$, and group weights optimization

$$\hat{\boldsymbol{p}}_n^* = \arg\min_{\boldsymbol{p} \in \mathcal{P}_G} \mathcal{L}_{\text{val}}(\hat{\boldsymbol{\theta}}_{n'}(\boldsymbol{p}), \boldsymbol{r}), \tag{10}$$

where the likelihood ratios $\boldsymbol{r}$ are given by $r_i = \frac{p_{\text{te}}(g_i)}{p_{\text{tr}}(g_i)}$. Also note that the group weights are restricted to the $G$-dimensional probability simplex $\mathcal{P}_G$. This normalization conveniently prevents modifying the effective learning rate of the optimization procedure (Ren et al., 2018).

We solve the bi-level optimization problem iteratively. We find $\hat{\boldsymbol{p}}_n^*$ in Equation 10 by means of exponentiated gradient descent (Nemirovsky & Yudin, 1983; Kivinen & Warmuth, 1997), where the parameter estimates $\hat{\boldsymbol{\theta}}_{n'}(\boldsymbol{p})$ are kept fixed. After each weight update, we re-calibrate the parameter estimates of Equation 1. Although convergence to the optimal group weights is not guaranteed due to potential non-convexity, gradient-based optimization of hyperparameters has previously been applied successfully (Pedregosa, 2016). Details of the optimization procedure are provided in Algorithm 1.

A step in this algorithm that requires extra attention is the computation of the gradient of the validation loss with respect to the group weights, commonly referred to as the *hyper-gradient*. For this, we use the chain rule

$$\nabla_{\boldsymbol{p}} \mathcal{L}_{\text{val}}(\hat{\boldsymbol{\theta}}_{n'}(\boldsymbol{p}), \boldsymbol{r}) = \nabla_{\hat{\boldsymbol{\theta}}_{n'}(\boldsymbol{p})} \mathcal{L}_{\text{val}}(\hat{\boldsymbol{\theta}}_{n'}(\boldsymbol{p}), \boldsymbol{r}) \cdot \nabla_{\boldsymbol{p}} \hat{\boldsymbol{\theta}}_{n'}(\boldsymbol{p})$$

and note that the first factor can typically be computed in a straightforward manner. For the gradient of the parameters with respect to the weights, we use the implicit function theorem. The idea of using the implicit function theorem for gradient-based optimization of hyperparameters was originally introduced by Bengio (2000). For the details of calculating the hyper-gradient, see Appendix A.

---

**Algorithm 1:** Estimation of the optimal weights $\hat{p}_n^*$ for GW-ERM

---

**Input:** data $\{y_i, g_i, \boldsymbol{x}_i\}_{i=1}^n$, training set size $n' < n$, maximum steps $T$, learning rate $\eta$, momentum $\gamma$ and likelihood ratios $\boldsymbol{r}$.

Initialize $\boldsymbol{p}_0$ such that $p_{0,g} = \frac{p_{\mathrm{te}}(g)}{p_{\mathrm{tr}}(g)}$ for all $g \in \mathcal{G}$. Normalize $p_{0,g} = \frac{p_{0,g}}{\sum_{g' \in \mathcal{G}} p_{0,g'}}$ for all $g \in \mathcal{G}$.

Split the data into a training set of size $n'$ and a validation set. Set $u_{0,g} = 0$ for all $g \in \mathcal{G}$.

**for** $t = 1, \ldots, T$ **do**

  Estimate $\hat{\boldsymbol{\theta}}_{n'}(\boldsymbol{p}_{t-1})$ as in Equation 1.
  Compute hyper-gradient $\boldsymbol{\zeta}_t = -\nabla_{\boldsymbol{p}_{t-1}} \mathcal{L}_{\mathrm{val}}(\hat{\boldsymbol{\theta}}_{n'}(\boldsymbol{p}_{t-1}), \boldsymbol{r})$ with implicit function theorem.
  Update weights $p_{t,g} = p_{t-1,g} \exp(\eta u_{t,g})$, with $u_{t,g} = \gamma u_{t-1,g} + (1 - \gamma)\zeta_{t,g}$, for all $g \in \mathcal{G}$.
  Normalize $p_{t,g} = \frac{p_{t,g}}{\sum_{g' \in \mathcal{G}} p_{t,g'}}$ for all $g \in \mathcal{G}$.

**end**

Return $\hat{\boldsymbol{p}}_n^* = \arg\min_{\boldsymbol{p}_t \in \{\boldsymbol{p}_0, \boldsymbol{p}_1, \ldots, \boldsymbol{p}_T\}} \mathcal{L}_{\mathrm{val}}(\hat{\boldsymbol{\theta}}_{n'}(\boldsymbol{p}_t), \boldsymbol{r})$.

---

## 5.2 Subsampling Large Groups

In the case of SUBG (Sagawa et al., 2020b), the weighted loss in Equation 1 is replaced by a training loss on a subset of the data created by subsampling without replacement. Let $\boldsymbol{s} = \{s_i^g | g \in \mathcal{G}, \ i = 1, \ldots, n_g\}$, where $s_i^g \in \{0, 1\}$ is a random variable denoting whether data point $i$ from group $g$ is included in the sample. The weights $v_g \in [0, 1]$ represent the fraction of data points to be subsampled, such that a total of $\lceil v_g n_g \rceil$ observations of group $g$ is drawn randomly and included in the training loss

$$\mathcal{L}_{\mathrm{train}}^{\mathrm{SUBG},n}(\boldsymbol{\theta}, \boldsymbol{v}) := \frac{1}{m} \sum_{g \in \mathcal{G}} \sum_{i=1}^{n_g} s_i^g \, \mathcal{L}(y_i^g, f_\theta(\boldsymbol{x}_i^g)), \tag{11}$$

where $y_i^g, \boldsymbol{x}_i^g$ are observations belonging to group $g$; $m = \sum_{g \in \mathcal{G}} \lceil v_g n_g \rceil$ and $\boldsymbol{v} = (v_1, v_2, \ldots, v_G)'$. This training loss is not differentiable with respect to $\boldsymbol{v}$, which poses a problem for the bi-level optimization procedure. This is resolved by using parameter estimates obtained by averaging over the random subsampling process,

$$\hat{\boldsymbol{\theta}}_n^{\mathrm{SUBG}}(\boldsymbol{v}) := \arg\min_{\boldsymbol{\theta} \in \Theta} \mathbb{E}_{\boldsymbol{s}} \left[ \mathcal{L}_{\mathrm{train}}^{\mathrm{SUBG},n}(\boldsymbol{\theta}, \boldsymbol{v}) \right] \approx \arg\min_{\boldsymbol{\theta} \in \Theta} \frac{1}{m} \sum_{g \in \mathcal{G}} \sum_{i=1}^{n_g} v_g \, \mathcal{L}(y_i^g, f_\theta(\boldsymbol{x}_i^g)), \tag{12}$$

where the approximation $\lceil v_g n_g \rceil / n_g \approx v_g$ was used. The optimized weights are still estimated with a weighted validation loss, $\hat{\boldsymbol{v}}_n^* = \arg\min_{\boldsymbol{v} \in \mathcal{V}_G} \mathcal{L}_{\mathrm{val}}(\hat{\boldsymbol{\theta}}_{n'}^{\mathrm{SUBG}}(\boldsymbol{v}), \boldsymbol{r})$, where $\mathcal{V}_G = \{\boldsymbol{v} \in [0, 1]^G \,|\, \exists g \in \mathcal{G} : v_g = 1\}$. Note that the weights $\boldsymbol{v}$ are not required to be normalized, but to avoid unnecessary loss of training data at least one of them must be equal to one. Apart from this and some minor differences in the computation of the hyper-gradient (see Appendix A), the bi-level optimization procedure for SUBG is similar to the one for GW-ERM as described in Algorithm 1.

A closely related method is DFR Kirichenko et al. (2023), in which an ensemble of models trained on subsampled groups is used to estimate the model parameters by averaging Equation 12. Typically, this is done on a validation set. Per model, the hyper-gradient is the same as for SUBG.

## 5.3 Group Distributional Robust Optimization

Instead of minimizing the overall expected loss with respect to a test distribution, GDRO aims to minimize the largest expected loss per group. This means that there is no separate test distribution. The associated training procedure minimizes a weighted sum of expected losses per group, where the loss weights $\boldsymbol{q} = (q_1, q_2, \ldots, q_G)'$ are updated during training and larger weights are given to groups with a larger expected loss (Oren et al., 2019; Sagawa et al., 2020a). We apply the framework of optimized importance weighting to the GDRO objective. As in Section 5.1, we work with group

weights $\boldsymbol{p}$. The model parameters are estimated via a weighted loss as in Equation 1, whereas the optimal weights are found by minimizing

$$\boldsymbol{p}_n^* = \underset{\boldsymbol{p} \in \mathcal{P}_G}{\arg\min} \, \mathbb{E}_{D_{\mathrm{tr}}^n \sim \mathcal{P}_{\mathrm{tr}}} \left[ \sup_{\boldsymbol{q} \in \mathcal{P}_G} \sum_{g=1}^{G} q_g \, \mathbb{E}_{(y, \boldsymbol{x}|g) \sim \mathcal{P}_{\mathrm{tr}}} \left[ \mathcal{L}(y, f_{\hat{\boldsymbol{\theta}}_n(\boldsymbol{p})}(\boldsymbol{x})) \, \Big| \, D_{\mathrm{tr}} \right] \right], \qquad (13)$$

where, as before, $\mathcal{P}_G$ is the $G$-dimensional probability simplex. The inner expectation of Equation 13 is estimated with a validation set that is withheld from the data used to train the model parameters. The bi-level optimization procedure now consists of updating three quantities per update step: the model parameters $\boldsymbol{\theta}$, the group weights $\boldsymbol{p}$ and the loss weights $\boldsymbol{q}$. See Algorithm 2 in Appendix D for details.

In the standard GDRO setting, the coefficients of the logistic regression are trained to minimize worst-group loss. This makes GDRO prone to overfitting (Sagawa et al., 2020a). We, instead, optimize the weights to minimize worst-group loss, whereas the model parameters are trained via Equation 1. In typical cases, because of the (much) lower dimension of the group weights, overfitting is less likely to be a problem, giving optimized weights an edge over standard GDRO.

### 5.4 GROUP INFERENCE METHODS

Acquiring group labels is often costly. Several methods (see Section 2) resolve this by estimating group membership. Methods in this realm, such as Just Train Twice (Liu et al., 2021), require a validation set with the ground-truth group labels to tune hyperparameters. A key hyperparameter of JTT is the extent to which misclassified observations should be upweighted, which is optimized via a grid search on a validation set. We replace this single upweighting factor by multiple group weights, leading to higher flexibility. Furthermore, we optimize them via gradient descent instead of grid search. Like JTT, we use estimated group labels instead of ground-truth group labels, and aim to find weights that minimize the worst-group loss.

## 6 EXPERIMENTS

The goal of this section is to verify empirically that existing state-of-the-art methods for addressing sub-population shifts benefit from using optimized group weights. We use benchmark classification datasets for studying sub-population shifts: two from computer vision (Waterbirds, CelebA) and one from natural language processing (MultiNLI). Groups are defined via a combination of the target label and a spurious attribute. The training and validation set are randomly split. For Waterbirds, this is different from previous implementations, where the validation set is balanced in terms of the groups (Sagawa et al., 2020a). We deviate from this, since it is more realistic that the training and validation sets are drawn from the same distribution. The distribution shift is established by measuring the generalization performance on a test set for which the groups have equal weights. For details on the datasets, including group sizes, see Appendix E.1.

We start by finetuning a pre-trained DNN on the respective datasets, a ResNet50 model (He et al., 2016) for Waterbirds and CelebA, and a BERT model (Devlin et al., 2019b) for MultiNLI. For five existing importance weighting methods that address sub-population shifts (GW-ERM, SUBG, DFR, GDRO, JTT), we then retrain the last layer of the DNN, both with the standard choice of weights prescribed by the method and the optimized weights as proposed in this paper. We focus on last-layer retraining, which amounts to training a logistic regression on the last-layer embeddings of the finetuned DNN, because of computational feasibility and because it has been shown to be sufficient for addressing sub-population shifts (Izmailov et al., 2022; Kirichenko et al., 2023). For each importance weighting method with standard weights choice, we use a grid search to optimize the respective hyperparameters, including an L1 penalty for sufficient regularization. To get a fair comparison, we use the same hyperparameters for optimized weights. This possibly gives an advantage to the standard choice of weights.

We repeat the above procedure, starting from a pre-trained DNN, five times for each dataset, importance weighting method and weights choice approach (standard vs. optimized). For each run, we use the test set to compute two metrics for generalization performance: weighted average accuracy, which is the equal-weight average of the accuracy per group, and worst-group accuracy.

We compare the difference in generalization performance between the standard weights choice and optimized weights via a one-sided $t$-test of paired samples. See Appendix E for details.

**Improving existing importance weighting methods**: Figure 2 compares the expected generalization performance of the standard weights choice and optimized weights. For a correct interpretation of the results, recall that for GW-ERM, SUBG and DFR, the objective of optimized weights is the weighted average accuracy, whereas for GDRO and JTT this is the worst-group accuracy. We observe that our approach consistently improves on the objective for which the weights are optimized, and is at least as good as standard weights with the exception of JTT on CelebA. Out of the 15 cases, we observe 12 times an increase that is statistically significant at the 10% level, and 8 times at the 5% level. Regarding the size of the improvements, in 5 cases there is an increase greater than 1%, with a maximum of 5.61%. To explain the case of JTT on CelebA, we note that even on the validation set optimizing weights did not improve the worst-group accuracy. This indicates that in this case our bi-level optimization procedure was unable to find better weights than the grid search of JTT.

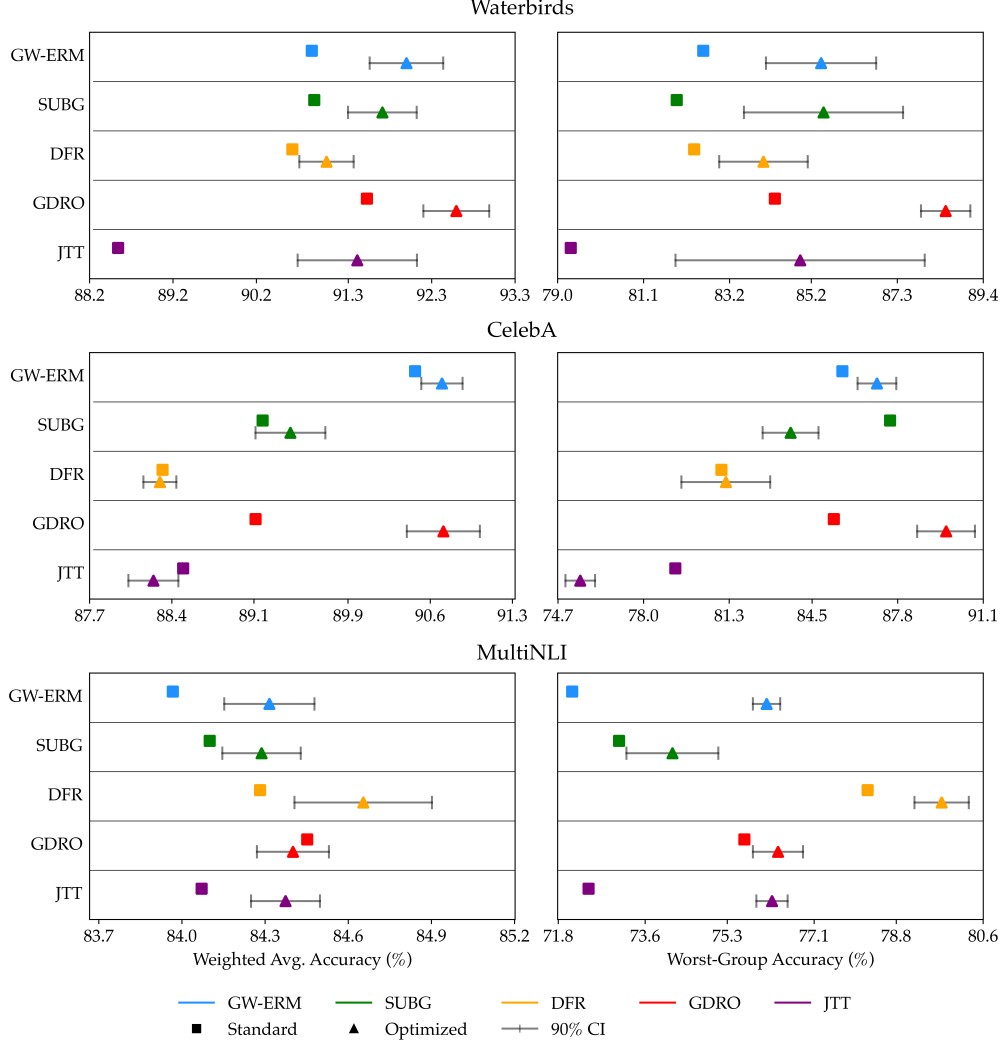

Figure 2: Estimated generalization performance (averaged over 5 runs) of standard weights choice (squares) versus optimized weights (triangles). Error bars reflect the paired-sample 90% confidence interval of the *difference*.

Furthermore, we find that optimizing weights often also improves the metric for which the weights were not optimized (worst-group accuracy for GW-ERM, SUBG and DFR; weighted average accuracy for GDRO and JTT). The only exception is SUBG on CelebA. In 10 cases, there is an improvement by more than 1%, with a maximum of 4.03%. This result was not necessarily expected, since

weighted average accuracy and group robustness are often opposed objectives (Agarwal et al., 2018; Martinez et al., 2021a). See appendix C for numerical details of the results.

**Role of the dataset size:** it can be observed in Figure 2 that the improvement due to optimized weights is less pronounced for CelebA and MultiNLI than for Waterbirds. One possible explanation is that the Waterbirds dataset is much smaller, leading to (very) small minority groups (see Appendix E.1). We have argued in Sections 3 and 4 that it is in particular this situation where the effect of optimized weights is expected to be large.

We test this for GW-ERM on CelebA in Figure 3, where we show the effect of optimized weights using only fractions of the original training and validation set (while keeping the ratio of the sizes the same). The improvement due to optimizing weights peaks when around 10% of the original data is used. Note that for smaller fractions of the data (5%) the effect decreases again. This is likely due to the validation set becoming (very) small, increasing the risk of overfitting on this data.

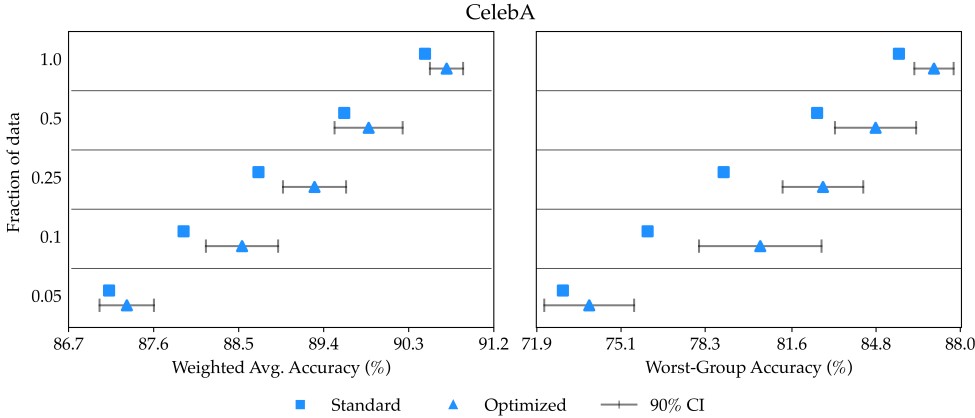

Figure 3: Estimated generalization performance (averaged over 5 runs) of standard weights choice (squares) versus optimized weights (triangles) for GW-ERM as a function of the sizes of the training and validation set. Error bars reflect the paired-sample 90% confidence interval of the *difference*.

**Robustness to other hyperparameters**: an additional benefit of optimizing weights is an improved robustness to the choice of the L1 penalty. We illustrate this in Figure 4 for GW-ERM and the Waterbirds dataset. Both generalization metrics vary less with optimized weights. We believe this is because optimized weights improve on the bias-variance trade-off, thereby compensating for suboptimal regularization choices in the hyperparameters.

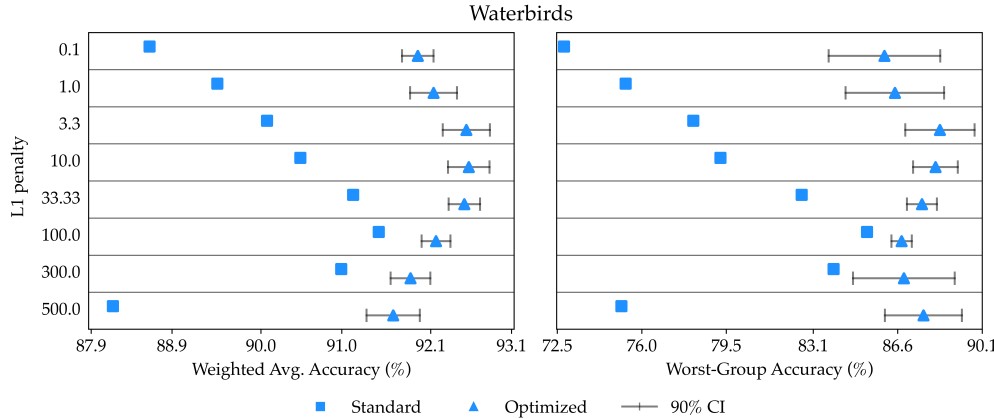

Figure 4: Estimated generalization performance (averaged over 5 runs) of standard weights choice (squares) versus optimized weights (triangles) for GW-ERM as a function of the L1 regularization parameter. Error bars reflect the paired-sample 90% confidence interval of the *difference*.

## 7 CONCLUSION AND DISCUSSION

In this paper we have shown that the standard choice of weights for importance weighting in the presence of sub-population shifts is often not optimal, due to the finite sample size of the training data. We have provided an estimation procedure for the optimal weights, which generally improves existing importance weighting methods in the case of last-layer retraining of DNNs.

Although last-layer retraining is known to be sufficient for addressing sub-population shifts (Izmailov et al., 2022; Kirichenko et al., 2023), it might be that retraining the whole DNN is the preferred option. In that case computational costs can become an issue, as calculating the hypergradient requires refitting the model on the training data and computing the inverted Hessian of the training loss with respect to the model parameters. To alleviate this, one might resort to several existing approaches to reduce computational costs for refitting the model (Maclaurin et al., 2015) or calculating the inverted Hessian (Lorraine et al., 2020).

The optimization properties of our method are another potential topic for future research. Although, with the exception of JTT on CelebA in Figure 2, the bi-level optimization procedure seems to converge well, formal convergence guarantees or the conditions under which the optimization in Equation 10 is convex were not investigated here.

The application to more challenging data-generating processes and models could lead to additional benefits of our approach. One example is the case in which the test distribution of interest is unknown (Sugiyama et al., 2012). Another example is concept bottleneck models, which are known to suffer from spurious correlations (Margeloiu et al., 2021; Heidemann et al., 2023). The existence of many concepts, and thus groups, leads to a severe reduction in the effective sample size. This is problematic for existing importance weighting techniques and gives our approach of optimizing weights the potential to lead to big improvements. We leave this for future work.

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

## A  CALCULATION OF THE HYPER-GRADIENT

The goal of this section is to provide further details on the calculation of the hyper-gradient, which is the gradient of the validation loss with respect to our hyperparameter $\boldsymbol{p}$, introduced in Section 5.1. This section should be read in conjunction with Section 5. Recall that the chain rule gives

$$\frac{\partial \mathcal{L}_{\text{val}}(\hat{\boldsymbol{\theta}}_{n'}(\boldsymbol{p}), \boldsymbol{r})}{\partial \boldsymbol{p}} = \frac{\partial \mathcal{L}_{\text{val}}(\hat{\boldsymbol{\theta}}_{n'}(\boldsymbol{p}), \boldsymbol{r})}{\partial \hat{\boldsymbol{\theta}}_{n'}(\boldsymbol{p})} \frac{\partial \hat{\boldsymbol{\theta}}_{n'}(\boldsymbol{p})}{\partial \boldsymbol{p}}. \tag{14}$$

Suppose we have a function $h(\boldsymbol{\theta}, \boldsymbol{p})$ which is twice differentiable, which will play the role of an (expected) training loss. Assume that the gradient of $h(\boldsymbol{\theta}, \boldsymbol{p})$ with respect to $\boldsymbol{\theta}$ is zero at the point $\hat{\boldsymbol{\theta}}_{n'}(\boldsymbol{p})$. We can then apply the implicit function theorem to this gradient at the point $\boldsymbol{\theta} = \hat{\boldsymbol{\theta}}_{n'}(\boldsymbol{p})$, and write

$$\frac{\partial \hat{\boldsymbol{\theta}}_{n'}(\boldsymbol{p})}{\partial \boldsymbol{p}} = - \left[ \frac{\partial^2 h(\boldsymbol{\theta}, \boldsymbol{p})}{\partial \boldsymbol{\theta} \partial \boldsymbol{\theta}^{\mathrm{T}}} \bigg|_{\boldsymbol{\theta} = \hat{\boldsymbol{\theta}}_{n'}(\boldsymbol{p})} \right]^{-1} \frac{\partial^2 h(\boldsymbol{\theta}, \boldsymbol{p})}{\partial \boldsymbol{\theta} \partial \boldsymbol{p}^{\mathrm{T}}} \bigg|_{\boldsymbol{\theta} = \hat{\boldsymbol{\theta}}_{n'}(\boldsymbol{p})}. \tag{15}$$

The gradient in Equation 15 depends on the form of $h$, and thus the estimation procedure of the parameters. Here, we discuss two such estimation procedures: (1) a group-weighted loss, and (2) subsampling per group without replacement.

**Group-weighted loss:** for simplicity of notation, assume that the group label $g$ takes values $1, 2, \ldots, G$. The group-weighted training loss is defined as

$$h(\boldsymbol{\theta}, \boldsymbol{p}) = \frac{1}{n} \left( \sum_{g=1}^{G} \frac{p_g}{p_{\text{tr},g}} \sum_{i=1}^{n_g} \mathcal{L}(y_i^g, f_\theta(\boldsymbol{x}_i^g)) \right). \tag{16}$$

The partial derivative of $h(\boldsymbol{\theta}, \boldsymbol{p})$ with respect to $p_g$ becomes a novel (weighted) loss function,

$$\frac{\partial h(\boldsymbol{\theta}, \boldsymbol{p})}{\partial p_g} = \frac{1}{n} \left( \frac{1}{p_{\text{tr},g}} \sum_{i=1}^{n_g} \mathcal{L}(y_i^g, f_\theta(\boldsymbol{x}_i^g)) - \frac{1}{(1 - \sum_{g'=1}^{G-1} p_{\text{tr},g'})} \sum_{i=1}^{n_G} \mathcal{L}(y_i^G, f_\theta(\boldsymbol{x}_i^G)) \right), \tag{17}$$

where we used that $p_G = 1 - \sum_{g'=1}^{G} p_{g'}$ due to normalization. Then, in order to determine $\frac{\partial^2 h(\boldsymbol{\theta}, \boldsymbol{p})}{\partial \boldsymbol{\theta} \partial \boldsymbol{p}^{\mathrm{T}}} \big|_{\boldsymbol{\theta} = \hat{\boldsymbol{\theta}}_{n'}(\boldsymbol{p})}$, we can calculate the derivative with respect to the estimated parameters for the novel loss function in Equation 17 for each $p_g$.

**Subsampling large groups**: here, we are interested in the hyper-gradient for the hyperparameter $\boldsymbol{v}$ instead of $\boldsymbol{p}$, as defined in Section 5.2. The parameters $\hat{\boldsymbol{\theta}}_{n'}(\boldsymbol{v})$ are not determined via a weighted loss, but rather by minimizing the training loss on a subsample of the data. However, the *expected* training loss can be rewritten as a weighted loss. To mimic the procedure of subsampling without replacement, we say that for each $g = 1, \ldots, G$ the random variables $s_i^g$ are identically distributed $\text{Bern}(v_g)$ random variables with constraint

$$\sum_{i=1}^{n_g} s_i^g = m_g, \text{ where } m_g = \lceil v_g n_g \rceil. \tag{18}$$

Recall from the main text that $0 \le v_g \le 1$, the fraction of samples used from the training data in our subsample. The expected training loss is then

$$\mathbb{E}_{\boldsymbol{s}} \left[ \mathcal{L}_{\text{train}}^{\text{SUBG},n}(\boldsymbol{\theta}, \boldsymbol{v}) \right] = \mathbb{E}_{\boldsymbol{s}} \left[ \frac{1}{m} \sum_{g=1}^{G} \sum_{i=1}^{n_g} s_i^g \mathcal{L}(y_i^g, f_\theta(\boldsymbol{x}_i^g)) \right] \tag{19}$$

$$= \frac{1}{m} \sum_{g=1}^{G} \sum_{i=1}^{n_g} \mathcal{L}(y_i^g, f_\theta(\boldsymbol{x}_i^g)) \frac{\lceil v_g n_g \rceil}{n_g} \tag{20}$$

where $m = \sum_{g=1}^{G} m_g$ is the total number of samples in our subsample. By fixing the value of each $m_g$ beforehand, $m$ is not stochastic. As explained in Section 5.2, we approximate this expected loss

via $\tilde{h}(\boldsymbol{\theta}, \boldsymbol{v}) = \frac{1}{m} \sum_{g=1}^{G} \sum_{i=1}^{n_g} v_g \, \mathcal{L}(y_i^g, f_\theta(\boldsymbol{x}_i^g))$. The partial derivative of $\tilde{h}(\boldsymbol{\theta}, \boldsymbol{v})$ with respect to $v_g$ again becomes a novel loss function

$$\frac{\partial \tilde{h}(\boldsymbol{\theta}, \boldsymbol{v})}{\partial v_g} = \frac{1}{m} \sum_{g=1}^{G} \sum_{i=1}^{n_g} \mathcal{L}(y_i^g, f_\theta(\boldsymbol{x}_i^g)). \tag{21}$$

Note the difference with Equation 17, because the $\boldsymbol{v}$ are not normalized. To determine $\frac{\partial^2 \tilde{h}(\boldsymbol{\theta}, \boldsymbol{v})}{\partial \boldsymbol{\theta} \partial \boldsymbol{v}^{\mathrm{T}}}\Big|_{\boldsymbol{\theta} = \hat{\boldsymbol{\theta}}_{n'}(\boldsymbol{v})}$, we can (again) calculate the derivative with respect to the estimated parameters for the novel loss function in Equation 21 for each $v_g$. In addition to the change in the calculation of the hyper-gradient, the only change to Algorithm 1 is that we do not normalize the weights $\boldsymbol{v}$.

# B  THEORETICAL ANALYSIS FOR LINEAR REGRESSION

The goal of this section is to derive the approximate expected loss given the data-generating process described in Section 4. It consists of four parts.

- In Section B.1 we give a description of how $\mathbb{E}_{D_{\mathrm{tr}}^n \sim \mathcal{P}_{\mathrm{tr}}} \left[ \mathbb{E}_{(y,\boldsymbol{x}) \sim \mathcal{P}_{\mathrm{te}}} [(y - \boldsymbol{x}^{\mathrm{T}} \hat{\boldsymbol{\beta}}(\boldsymbol{w}))^2 | D_{\mathrm{tr}}] \right]$ can be decomposed into a bias and variance term in the case of linear regression, and data stemming from $G$ groups.

- In Section B.2 we provide the approximations of the OLS coefficients for large but finite $n$, given the data-generating process described in Section 4.

- In Section B.3, combining the results of the previous two sections, we derive the (approximate) bias-variance decomposition for our data-generating process.

- Finally, in Section B.4 we provide a numerical comparison between simulations and our approximation.

## B.1  THE BIAS-VARIANCE TRADEOFF

For clarity, throughout the next three sections we will use the subscript tr to indicate if random variables are drawn from the training distribution, and the subscript te that they are drawn from the test distribution. We use the mean-squared error as the loss function, and use a linear regression where the coefficients depend on our weights.

$$\mathbb{E}_{D_{\mathrm{tr}}^n \sim \mathcal{P}_{\mathrm{tr}}} \left[ \mathbb{E}_{y_{\mathrm{te}}, \boldsymbol{x}_{\mathrm{te}}} [(y_{\mathrm{te}} - f_{\hat{\boldsymbol{\theta}}_n(\boldsymbol{w})}(\boldsymbol{x}))^2 | D_{\mathrm{tr}}] \right] = \mathbb{E}_{D_{\mathrm{tr}}^n \sim \mathcal{P}_{\mathrm{tr}}} \left[ \mathbb{E}_{y_{\mathrm{te}}, \boldsymbol{x}_{\mathrm{te}}} [(y_{\mathrm{te}} - \boldsymbol{x}_{\mathrm{te}}^{\mathrm{T}} \hat{\boldsymbol{\beta}}(\boldsymbol{w}))^2 | D_{\mathrm{tr}}] \right].$$

The following only requires the assumptions stated in Section 4 regarding the independence of $\boldsymbol{x}_{\mathrm{te}}$ from $\epsilon_{\mathrm{te}}$. The expected mean squared error over a test distribution, conditional on the data coming from group $g$, can be written as

$$\mathbb{E}_{D_{\mathrm{tr}}^n \sim \mathcal{P}_{\mathrm{tr}}} \left[ \mathbb{E}_{\boldsymbol{x}_{\mathrm{te}}, y_{\mathrm{te}}} [(y_{\mathrm{te}} - \boldsymbol{x}_{\mathrm{te}}^{\mathrm{T}} \hat{\boldsymbol{\beta}}(\boldsymbol{w}))^2 | g, D_{\mathrm{tr}}] \right]$$

$$= \mathbb{E}_{D_{\mathrm{tr}}^n \sim \mathcal{P}_{\mathrm{tr}}} \left[ \mathbb{E}_{\boldsymbol{x}_{\mathrm{te}}, y_{\mathrm{te}}} [y_{\mathrm{te}}^2 - 2 y_{\mathrm{te}} \boldsymbol{x}_{\mathrm{te}}^{\mathrm{T}} \hat{\boldsymbol{\beta}}(\boldsymbol{w}) + (\boldsymbol{x}_{\mathrm{te}}^{\mathrm{T}} \hat{\boldsymbol{\beta}}(\boldsymbol{w}))^2 | g, D_{\mathrm{tr}}] \right]$$

$$= \mathbb{E}_{y_{\mathrm{te}}} [y_{\mathrm{te}}^2 | g] - 2 \mathbb{E}_{D_{\mathrm{tr}}^n \sim \mathcal{P}_{\mathrm{tr}}} \left[ \mathbb{E}_{\boldsymbol{x}_{\mathrm{te}}, y_{\mathrm{te}}} \left[ y_{\mathrm{te}} \boldsymbol{x}_{\mathrm{te}}^{\mathrm{T}} \hat{\boldsymbol{\beta}}(\boldsymbol{w}) | g, D_{\mathrm{tr}} \right] \right]$$

$$+ \mathbb{E}_{D_{\mathrm{tr}}^n \sim \mathcal{P}_{\mathrm{tr}}} \left[ \mathbb{E}_{p(\boldsymbol{x}_{\mathrm{te}})} \left[ (\boldsymbol{x}_{\mathrm{te}}^{\mathrm{T}} \hat{\boldsymbol{\beta}}(\boldsymbol{w}))^2 | g, D_{\mathrm{tr}} \right] \right].$$

For each group $g$, the following holds:

$$\mathbb{E}_{y_{\text{te}}}[y_{\text{te}}^2|g] = \mathbb{E}_{\boldsymbol{x}_{\text{te}}}[(\boldsymbol{x}_{\text{te}}^{\text{T}}\boldsymbol{\beta}^g + \epsilon_{\text{te}})^2|g] = \mathbb{E}_{\boldsymbol{x}_{\text{te}}}[(\boldsymbol{x}_{\text{te}}^{\text{T}}\boldsymbol{\beta}^g)^2|g] + \sigma^2,$$

$$\mathbb{E}_{D_{\text{tr}}^n \sim \mathcal{P}_{\text{tr}}}\left[\mathbb{E}_{\boldsymbol{x}_{\text{te}},y_{\text{te}}}\left[y\boldsymbol{x}_{\text{te}}^{\text{T}}\hat{\boldsymbol{\beta}}(\boldsymbol{w})|g, D_{\text{tr}}\right]\right]$$

$$= \mathbb{E}_{D_{\text{tr}}^n \sim \mathcal{P}_{\text{tr}}}\left[\mathbb{E}_{\boldsymbol{x}_{\text{te}},y_{\text{te}}}\left[(\boldsymbol{x}_{\text{te}}^{\text{T}}\boldsymbol{\beta}^g + \epsilon_{\text{te}})\boldsymbol{x}_{\text{te}}^{\text{T}}\hat{\boldsymbol{\beta}}(\boldsymbol{w})|g, D_{\text{tr}}\right]\right]$$

$$= \mathbb{E}_{\boldsymbol{x}_{\text{te}}}[\boldsymbol{x}_{\text{te}}^{\text{T}}\boldsymbol{\beta}^g|g]\mathbb{E}_{D_{\text{tr}}^n \sim \mathcal{P}_{\text{tr}}}\left[\mathbb{E}_{\boldsymbol{x}_{\text{te}}}\left[\boldsymbol{x}_{\text{te}}^{\text{T}}\hat{\boldsymbol{\beta}}(\boldsymbol{w})|g, D_{\text{tr}}\right]\right],$$

$$\mathbb{E}_{D_{\text{tr}}^n \sim \mathcal{P}_{\text{tr}}}\left[\mathbb{E}_{\boldsymbol{x}_{\text{te}},y_{\text{te}}}\left[(\boldsymbol{x}_{\text{te}}^{\text{T}}\hat{\boldsymbol{\beta}}(\boldsymbol{w}))^2|g, D_{\text{tr}}\right]\right]$$

$$= \mathbb{E}_{D_{\text{tr}}^n \sim \mathcal{P}_{\text{tr}}}\left[\text{Var}\left(\boldsymbol{x}_{\text{te}}^{\text{T}}\hat{\boldsymbol{\beta}}(\boldsymbol{w})|g, D_{\text{tr}}\right) + \mathbb{E}_{\boldsymbol{x}_{\text{te}}}\left[\boldsymbol{x}_{\text{te}}^{\text{T}}\hat{\boldsymbol{\beta}}(\boldsymbol{w})|g, D_{\text{tr}}\right]^2|D_{\text{tr}}\right].$$

We can then write the expected loss conditional on $g$ as:

$$\mathbb{E}_{D_{\text{tr}}^n \sim \mathcal{P}_{\text{tr}}}\left[\mathbb{E}_{\boldsymbol{x}_{\text{te}},y_{\text{te}}}[(y_{\text{te}} - \boldsymbol{x}_{\text{te}}^{\text{T}}\hat{\boldsymbol{\beta}}(\boldsymbol{w}))^2|g, D_{\text{tr}}]\right]$$

$$= \mathbb{E}_{\boldsymbol{x}_{\text{te}}}[(\boldsymbol{x}_{\text{te}}^{\text{T}}\boldsymbol{\beta}^g)^2|g] - 2\mathbb{E}_{\boldsymbol{x}_{\text{te}}}[\boldsymbol{x}_{\text{te}}^{\text{T}}\boldsymbol{\beta}^g|g]\mathbb{E}_{D_{\text{tr}}^n \sim \mathcal{P}_{\text{tr}}}\left[\mathbb{E}_{\boldsymbol{x}_{\text{te}}}\left[\boldsymbol{x}_{\text{te}}^{\text{T}}\hat{\boldsymbol{\beta}}(\boldsymbol{w})|g, D_{\text{tr}}\right]\right]$$

$$+ \mathbb{E}_{D_{\text{tr}}^n \sim \mathcal{P}_{\text{tr}}}\left[\mathbb{E}_{\boldsymbol{x}_{\text{te}}}\left[\boldsymbol{x}_{\text{te}}^{\text{T}}\hat{\boldsymbol{\beta}}(\boldsymbol{w})|g, D_{\text{tr}}\right]^2|D_{\text{tr}}\right]$$

$$+ \mathbb{E}_{D_{\text{tr}}^n \sim \mathcal{P}_{\text{tr}}}\left[\text{Var}\left(\boldsymbol{x}_{\text{te}}^{\text{T}}\hat{\boldsymbol{\beta}}(\boldsymbol{w})|g, D_{\text{tr}}\right)\Big|D_{\text{tr}}\right] + \sigma^2.$$

Suppose we write this expected loss conditional on both $\boldsymbol{x}_{\text{te}}$ and $g$. Then, it can be written as:

$$\mathbb{E}_{D_{\text{tr}}^n \sim \mathcal{P}_{\text{tr}}}\left[\mathbb{E}_{y_{\text{te}}}[(y_{\text{te}} - \boldsymbol{x}_{\text{te}}^{\text{T}}\hat{\boldsymbol{\beta}}(\boldsymbol{w}))^2|\boldsymbol{x}_{\text{te}}, g]\right]$$

$$= (\boldsymbol{x}_{\text{te}}^{\text{T}}\boldsymbol{\beta}^g - \boldsymbol{x}_{\text{te}}^{\text{T}}\mathbb{E}_{D_{\text{tr}}^n \sim \mathcal{P}_{\text{tr}}}\left[\hat{\boldsymbol{\beta}}(\boldsymbol{w})\right])^2 + \boldsymbol{x}_{\text{te}}^{\text{T}}\mathbb{E}_{D_{\text{tr}}^n \sim \mathcal{P}_{\text{tr}}}\left[\text{Var}(\hat{\boldsymbol{\beta}}(\boldsymbol{w})|D_{\text{tr}})\right]\boldsymbol{x}_{\text{te}} + \sigma_g^2.$$

To further specify the expected loss, we will use the data-generating process specified in Section 4. Most notably, we will use that the $\boldsymbol{x}_{\text{te}}$ does not change with the group $g$, that there are only two groups, and our definition of the coefficients per group.

By the law of total probability, the expected loss conditional on only $\boldsymbol{x}_{\text{te}}$ can be written as a sum of the expected losses conditional on $\boldsymbol{x}_{\text{te}}, g$.

$$\mathbb{E}_{D_{\text{tr}}^n \sim \mathcal{P}_{\text{tr}}}\left[\mathbb{E}_{y_{\text{te}},g_{\text{te}}}[(y_{\text{te}} - \boldsymbol{x}_{\text{te}}^{\text{T}}\hat{\boldsymbol{\beta}}(\boldsymbol{w}))^2|\boldsymbol{x}_{\text{te}}]\right]$$

$$= \sum_{g=1}^{G} p(g_{\text{te}} = g)\mathbb{E}_{D_{\text{tr}}^n \sim \mathcal{P}_{\text{tr}}}\left[\mathbb{E}_{y_{\text{te}}}[(y_{\text{te}} - \boldsymbol{x}_{\text{te}}^{\text{T}}\hat{\boldsymbol{\beta}}(\boldsymbol{w}))^2|\boldsymbol{x}_{\text{te}}, g]\right]$$

$$= p_{\text{te}}((\boldsymbol{x}_{\text{te}}^{\text{T}}\boldsymbol{\beta}^1 - \boldsymbol{x}_{\text{te}}^{\text{T}}\mathbb{E}_{D_{\text{tr}}^n \sim \mathcal{P}_{\text{tr}}}[\hat{\boldsymbol{\beta}}(\boldsymbol{w})])^2 + (1 - p_{\text{te}})((\boldsymbol{x}_{\text{te}}^{\text{T}}\boldsymbol{\beta}^0 - \boldsymbol{x}_{\text{te}}^{\text{T}}\mathbb{E}_{D_{\text{tr}}^n \sim \mathcal{P}_{\text{tr}}}[\hat{\boldsymbol{\beta}}(\boldsymbol{w})])^2$$

$$+ \boldsymbol{x}_{\text{te}}^{\text{T}}\mathbb{E}_{D_{\text{tr}}^n \sim \mathcal{P}_{\text{tr}}}\left[\text{Var}(\hat{\boldsymbol{\beta}}(\boldsymbol{w})|D_{\text{tr}})\right]\boldsymbol{x}_{\text{te}} + \sigma^2. \tag{22}$$

## B.2 THE ASYMPTOTIC BIAS AND VARIANCE OF THE COEFFICIENTS

The goal of this subsection is to define the expectation and variance of the WLS estimator as defined per equation 7. Because we have fixed the amount of observations per group, we can state the following

$$\frac{n_1}{n} = \frac{np_{\text{tr}}}{n} = p_{\text{tr}}, \quad \frac{n_0}{n} = \frac{n(1 - p_{\text{tr}})}{n} = (1 - p_{\text{tr}}), \tag{23}$$

$$w_1\frac{n_1}{n} = \frac{p}{p_{\text{tr}}}p_{\text{tr}} = p, \quad w_0\frac{n_0}{n} = \frac{(1 - p)}{(1 - p_{\text{tr}})}(1 - p_{\text{tr}}) = (1 - p). \tag{24}$$

This means we can write

$$\boldsymbol{X}_{\text{tr}}^{\text{T}}\boldsymbol{W}\boldsymbol{X}_{\text{tr}} = w_1\boldsymbol{X}_{\text{tr},1}^{\text{T}}\boldsymbol{X}_{\text{tr},1} + w_0\boldsymbol{X}_{\text{tr},0}^{\text{T}}\boldsymbol{X}_{\text{tr},0}, \quad \boldsymbol{X}_{\text{tr}}^{\text{T}}\boldsymbol{W}\boldsymbol{y}_{\text{tr}} = w_1\boldsymbol{X}_{\text{tr},1}^{\text{T}}\boldsymbol{y}_1 + w_0\boldsymbol{X}_{\text{tr},0}^{\text{T}}\boldsymbol{y}_0, \tag{25}$$

and similarly $\boldsymbol{X}_{\mathrm{tr}}^{\mathrm{T}} \boldsymbol{W} \boldsymbol{W} \boldsymbol{X}_{\mathrm{tr}} = w_1^2 \boldsymbol{X}_{\mathrm{tr},1}^{\mathrm{T}} \boldsymbol{X}_{\mathrm{tr},1} + w_0^2 \boldsymbol{X}_{\mathrm{tr},0}^{\mathrm{T}} \boldsymbol{X}_{\mathrm{tr},0}$, where $\boldsymbol{X}_{\mathrm{tr},1} \in \mathbb{R}^{n_1 \times d}, \boldsymbol{X}_{\mathrm{tr},0} \in \mathbb{R}^{n_0 \times d}$ are matrices where for the observations holds that $g_i = 1, g_i = 0$ respectively, and similarly $\boldsymbol{y}_1 \in \mathbb{R}^{n_1}, \boldsymbol{y}_0 \in \mathbb{R}^{n_0}$.

In the following analysis we will use that

$$
\begin{aligned}
\left(\frac{1}{n}(w_1 \boldsymbol{X}_{\mathrm{tr},1}^{\mathrm{T}} \boldsymbol{X}_{\mathrm{tr},1} + w_0 \boldsymbol{X}_{\mathrm{tr},0}^{\mathrm{T}} \boldsymbol{X}_{\mathrm{tr},0})\right)^{-1} &= \left(\frac{1}{n}(w_1 \frac{n_1}{n_1} \boldsymbol{X}_{\mathrm{tr},1}^{\mathrm{T}} \boldsymbol{X}_{\mathrm{tr},1} + w_0 \frac{n_0}{n_0} \boldsymbol{X}_{\mathrm{tr},0}^{\mathrm{T}} \boldsymbol{X}_{\mathrm{tr},0})\right)^{-1} \\
&= \left(p \frac{1}{n_1} \boldsymbol{X}_{\mathrm{tr},1}^{\mathrm{T}} \boldsymbol{X}_{\mathrm{tr},1} + (1-p) \frac{1}{n_0} \boldsymbol{X}_{\mathrm{tr},0}^{\mathrm{T}} \boldsymbol{X}_{\mathrm{tr},0}\right)^{-1}.
\end{aligned}
\tag{26}
$$

Similarly,

$$
\begin{aligned}
\frac{1}{n}\left(w_1^2 \boldsymbol{X}_{\mathrm{tr},1}^{\mathrm{T}} \boldsymbol{X}_{\mathrm{tr},1} + w_0^2 \boldsymbol{X}_{\mathrm{tr},0}^{\mathrm{T}} \boldsymbol{X}_{\mathrm{tr},0}\right) &= \frac{1}{n}\left(w_1^2 \frac{n_1}{n_1} \boldsymbol{X}_{\mathrm{tr},1}^{\mathrm{T}} \boldsymbol{X}_{\mathrm{tr},1} + w_0^2 \frac{n_0}{n_0} \boldsymbol{X}_{\mathrm{tr},0}^{\mathrm{T}} \boldsymbol{X}_{\mathrm{tr},0}\right) \\
&= \left(\frac{p^2}{p_{\mathrm{tr}}} \frac{1}{n_1} \boldsymbol{X}_{\mathrm{tr},1}^{\mathrm{T}} \boldsymbol{X}_{\mathrm{tr},1} + \frac{(1-p)^2}{(1-p_{\mathrm{tr}})} \frac{1}{n_0} \boldsymbol{X}_{\mathrm{tr},0}^{\mathrm{T}} \boldsymbol{X}_{\mathrm{tr},0}\right). \tag{27}
\end{aligned}
$$

Finally, we will use that

$$
\begin{aligned}
\frac{1}{n}(w_1 \boldsymbol{X}_{\mathrm{tr},1}^{\mathrm{T}} \boldsymbol{y}_1 + w_0 \boldsymbol{X}_{\mathrm{tr},0}^{\mathrm{T}} \boldsymbol{y}_0) &= \frac{1}{n}(w_1 \frac{n_1}{n_1} \boldsymbol{X}_{\mathrm{tr},1}^{\mathrm{T}} \boldsymbol{y}_1 + w_0 \frac{n_0}{n_0} \boldsymbol{X}_{\mathrm{tr},0}^{\mathrm{T}} \boldsymbol{y}_0) \\
&= p \frac{1}{n_1} \boldsymbol{X}_{\mathrm{tr},1}^{\mathrm{T}} \boldsymbol{y}_1 + (1-p) \frac{1}{n_0} \boldsymbol{X}_{\mathrm{tr},0}^{\mathrm{T}} \boldsymbol{y}_0. \tag{28}
\end{aligned}
$$

Using Equations 26 and 28, the WLS estimator can be re-written as

$$
\begin{aligned}
(\boldsymbol{X}_{\mathrm{tr}}^{\mathrm{T}} \boldsymbol{W} \boldsymbol{X}_{\mathrm{tr}})^{-1} \boldsymbol{X}_{\mathrm{tr}}^{\mathrm{T}} \boldsymbol{W} \boldsymbol{y}_{\mathrm{tr}} &= (\frac{1}{n} \boldsymbol{X}_{\mathrm{tr}}^{\mathrm{T}} \boldsymbol{W} \boldsymbol{X}_{\mathrm{tr}})^{-1} \frac{1}{n} \boldsymbol{X}_{\mathrm{tr}}^{\mathrm{T}} \boldsymbol{W} \boldsymbol{y}_{\mathrm{tr}} \\
&= \left(\frac{1}{n} \boldsymbol{X}_{\mathrm{tr}}^{\mathrm{T}} \boldsymbol{W} \boldsymbol{X}_{\mathrm{tr}}\right)^{-1} \left(p \frac{1}{n_1} \boldsymbol{X}_{\mathrm{tr},1}^{\mathrm{T}} \boldsymbol{y}_1 + (1-p) \frac{1}{n_0} \boldsymbol{X}_{\mathrm{tr},0}^{\mathrm{T}} \boldsymbol{y}_0\right).
\end{aligned}
$$

The expectation of this estimator, conditional on $\boldsymbol{g}, \boldsymbol{X}$ becomes

$$
\begin{aligned}
\mathbb{E}[\hat{\boldsymbol{\beta}}(p)|\boldsymbol{g}_{\mathrm{tr}}, \boldsymbol{X}_{\mathrm{tr}}] &= \mathbb{E}\left[(\frac{1}{n} \boldsymbol{X}_{\mathrm{tr}}^{\mathrm{T}} \boldsymbol{W} \boldsymbol{X}_{\mathrm{tr}})^{-1} \frac{1}{n} \boldsymbol{X}_{\mathrm{tr}}^{\mathrm{T}} \boldsymbol{W} \boldsymbol{y}_{\mathrm{tr}}|\boldsymbol{g}_{\mathrm{tr}}, \boldsymbol{X}_{\mathrm{tr}}\right] \\
&= \mathbb{E}\left[\left(\frac{1}{n} \boldsymbol{X}_{\mathrm{tr}}^{\mathrm{T}} \boldsymbol{W} \boldsymbol{X}_{\mathrm{tr}}\right)^{-1} \left(p \frac{1}{n_1} \boldsymbol{X}_{\mathrm{tr},1}^{\mathrm{T}} \boldsymbol{y}_1 + (1-p) \frac{1}{n_0} \boldsymbol{X}_{\mathrm{tr},0}^{\mathrm{T}} \boldsymbol{y}_0\right) |\boldsymbol{g}_{\mathrm{tr}}, \boldsymbol{X}_{\mathrm{tr}}\right] \\
&= \left(\frac{1}{n} \boldsymbol{X}_{\mathrm{tr}}^{\mathrm{T}} \boldsymbol{W} \boldsymbol{X}_{\mathrm{tr}}\right)^{-1} \left(p \frac{1}{n_1} \boldsymbol{X}_{\mathrm{tr},1}^{\mathrm{T}} \mathbb{E}[\boldsymbol{y}_1|\boldsymbol{X}] + (1-p) \frac{1}{n_0} \boldsymbol{X}_{\mathrm{tr},0}^{\mathrm{T}} \mathbb{E}[\boldsymbol{y}_0|\boldsymbol{X}]\right) \\
&= \left(\frac{1}{n} \boldsymbol{X}_{\mathrm{tr}}^{\mathrm{T}} \boldsymbol{W} \boldsymbol{X}_{\mathrm{tr}}\right)^{-1} \left(p \frac{1}{n_1} \boldsymbol{X}_{\mathrm{tr},1}^{\mathrm{T}} \boldsymbol{X}_{\mathrm{tr},1} \boldsymbol{\beta}^1 + (1-p) \frac{1}{n_0} \boldsymbol{X}_{\mathrm{tr},0}^{\mathrm{T}} \boldsymbol{X}_{\mathrm{tr},0} \boldsymbol{\beta}^0\right).
\end{aligned}
$$

Given our DGP we can write the variance of $\boldsymbol{y}_{\mathrm{tr}}$ conditional on $\boldsymbol{g}_{\mathrm{tr}}, \boldsymbol{X}_{\mathrm{tr}}$ as

$$
\mathrm{Var}(\boldsymbol{y}_{\mathrm{tr}}|\boldsymbol{g}_{\mathrm{tr}}, \boldsymbol{X}_{\mathrm{tr}}) = \sigma^2 \boldsymbol{I}. \tag{29}
$$

We can then define the variance of the WLS estimator conditional on $\boldsymbol{g}, \boldsymbol{X}$:

$$
\begin{aligned}
\mathrm{Var}(\hat{\boldsymbol{\beta}}(p)|\boldsymbol{g}_{\mathrm{tr}}, \boldsymbol{X}_{\mathrm{tr}}) &= \mathrm{Var}((\boldsymbol{X}_{\mathrm{tr}}^{\mathrm{T}} \boldsymbol{W} \boldsymbol{X}_{\mathrm{tr}})^{-1} \boldsymbol{X}_{\mathrm{tr}}^{\mathrm{T}} \boldsymbol{W} \boldsymbol{y}_{\mathrm{tr}}|\boldsymbol{g}_{\mathrm{tr}}, \boldsymbol{X}_{\mathrm{tr}}) \\
&= (\boldsymbol{X}_{\mathrm{tr}}^{\mathrm{T}} \boldsymbol{W} \boldsymbol{X}_{\mathrm{tr}})^{-1} \boldsymbol{X}_{\mathrm{tr}}^{\mathrm{T}} \boldsymbol{W} \mathrm{Var}(\boldsymbol{y}_{\mathrm{tr}}|\boldsymbol{g}_{\mathrm{tr}}, \boldsymbol{X}_{\mathrm{tr}}) \boldsymbol{W} \boldsymbol{X}_{\mathrm{tr}} (\boldsymbol{X}_{\mathrm{tr}}^{\mathrm{T}} \boldsymbol{W} \boldsymbol{X}_{\mathrm{tr}})^{-1} \\
&= \sigma^2 (\boldsymbol{X}_{\mathrm{tr}}^{\mathrm{T}} \boldsymbol{W} \boldsymbol{X}_{\mathrm{tr}})^{-1} \boldsymbol{X}_{\mathrm{tr}}^{\mathrm{T}} \boldsymbol{W} \boldsymbol{W} \boldsymbol{X}_{\mathrm{tr}} (\boldsymbol{X}_{\mathrm{tr}}^{\mathrm{T}} \boldsymbol{W} \boldsymbol{X}_{\mathrm{tr}})^{-1} \tag{30}
\end{aligned}
$$

Similar to the data-generating process in Section 4, we assume here that each $x$ follows the same distribution regardless of the group. Given this, we can state $\mathbb{E}[\frac{1}{n_1}X_{\text{tr},1}^{\text{T}}X_{\text{tr},1}] = \mathbb{E}[\frac{1}{n_0}X_{\text{tr},0}^{\text{T}}X_{\text{tr},0}] = \Sigma$, e.g. they have the same expectation. Assuming $\Sigma$ is a positive definite matrix with finite elements, $\frac{1}{n_1}X_1^{\text{T}}X_1, \frac{1}{n_0}X_0^{\text{T}}X_0$ converge in probability to it, e.g.

$$\operatorname*{plim}_{n_1 \to \infty} \frac{1}{n_1}X_1^{\text{T}}X_1 = \operatorname*{plim}_{n_0 \to \infty}\frac{1}{n_0}X_0^{\text{T}}X_0 = \Sigma. \tag{31}$$

Since we assume each of our observations of $x$ are identically and independently distributed (IID), we can write

$$\frac{1}{n_1}X_{\text{tr},1}^{\text{T}}X_{\text{tr},1} = \frac{1}{n_0}X_{\text{tr},0}^{\text{T}}X_{\text{tr},0} = \Sigma + \mathcal{O}_p(\frac{1}{\sqrt{n}}). \tag{32}$$

See Equation 14.4-7 of Bishop et al. (2007) for a general version of this result. We can thus write

$$\left(p\frac{1}{n_1}X_{\text{tr},1}^{\text{T}}X_{\text{tr},1} + (1-p)\frac{1}{n_0}X_{\text{tr},0}^{\text{T}}X_{\text{tr},0}\right) = p\Sigma + (1-p)\Sigma + \mathcal{O}_p(\frac{1}{\sqrt{n}}) = \Sigma + \mathcal{O}_p(\frac{1}{\sqrt{n}}),$$

and also,

$$\left(p\frac{1}{n_1}X_{\text{tr},1}^{\text{T}}X_{\text{tr},1} + (1-p)\frac{1}{n_0}X_{\text{tr},0}^{\text{T}}X_{\text{tr},0}\right)^{-1} = \left(\Sigma + \mathcal{O}_p(\frac{1}{\sqrt{n}})\right)^{-1}$$
$$= \Sigma^{-1} - \Sigma^{-1}\mathcal{O}_p(\frac{1}{\sqrt{n}})\Sigma^{-1}$$
$$= \Sigma^{-1} - \mathcal{O}_p(\frac{1}{\sqrt{n}}).$$

The $\mathcal{O}_p(\frac{1}{\sqrt{n}})$ indicates $\frac{1}{n_1}X_{\text{tr},1}^{\text{T}}X_{\text{tr},1}, \frac{1}{n_0}X_{\text{tr},1}^{\text{T}}X_{\text{tr},1}$ are bounded in probability (see van der Vaart (1998) Section 2.2). Thus, for a large $n$, we can approximate $\mathbb{E}\left[\left(X_{\text{tr}}^{\text{T}}WX_{\text{tr}}\right)^{-1}\right]$ using $\Sigma^{-1}$.

We use the following approximation

$$\mathbb{E}_{D_{\text{tr}}^n \sim \mathcal{P}_{\text{tr}}}[\hat{\beta}(w) \tag{33}$$
$$= \mathbb{E}_{g_{\text{tr}}, X_{\text{tr}}}\left[\mathbb{E}\left[\left(X_{\text{tr}}^{\text{T}}WX_{\text{tr}}\right)^{-1}\left(p\frac{1}{n_1}X_{\text{tr},1}^{\text{T}}X_{\text{tr},1}\beta^1 + (1-p)\frac{1}{n_0}X_{\text{tr},0}^{\text{T}}X_{\text{tr},0}\beta^0\right)\right]\right]$$
$$= \Sigma^{-1}(p\Sigma\beta^1 + (1-p)\Sigma\beta^0) + \mathcal{O}_p(\frac{1}{\sqrt{n}})$$
$$\approx p\beta^1 + (1-p)\beta^0. \tag{34}$$

Similarly, we can define an approximation for the conditional variance. First, we define

$$\operatorname*{plim}_{n \to \infty}\frac{1}{n}X_{\text{tr}}^{\text{T}}WWX_{\text{tr}} = \left(\frac{p^2}{p_{\text{tr}}} + \frac{(1-p)^2}{(1-p_{\text{tr}})}\right)\Sigma. \tag{35}$$

We can now state

$$\mathbb{E}_{D_{\text{tr}}^n \sim \mathcal{P}_{\text{tr}}}\left[\text{Var}(\hat{\beta}(w)|D_{\text{tr}})\right] = \mathbb{E}_{g_{\text{tr}}, X_{\text{tr}}}\left[\sigma^2(X_{\text{tr}}^{\text{T}}WX_{\text{tr}})^{-1}X_{\text{tr}}^{\text{T}}WWX_{\text{tr}}(X_{\text{tr}}^{\text{T}}WX_{\text{tr}})^{-1}\right]$$
$$\approx \sigma^2\left(\frac{p^2}{p_{\text{tr}}} + \frac{(1-p)^2}{(1-p_{\text{tr}})}\right)\Sigma^{-1}. \tag{36}$$

### B.3 THE EXPECTED LOSS

If we substitute in our approximations derived in Equations 34,36 in Equation 22, we get

$$\mathbb{E}_{D_{\text{tr}}^n \sim \mathcal{P}_{\text{tr}}}\left[\mathbb{E}_{p_{\text{te}}(y)}[(y - x_{\text{te}}^{\text{T}}\hat{\beta}(w))^2|x_{\text{te}}]\right] \approx \left(\sum_{g=1}^{G}p_{\text{te},g}((x_{\text{te}}^{\text{T}}\beta^g - x_{\text{te}}^{\text{T}}(p\beta^1 + (1-p)\beta^0))^2 + \sigma_g^2\right)$$
$$+ x_{\text{te}}^{\text{T}}\frac{1}{n}\sigma^2\left(\frac{p^2}{p_{\text{tr},1}} + \frac{(1-p)^2}{(1-p_{\text{tr},1})}\right)\Sigma^{-1}x_{\text{te}}.$$

Now, we will define the expected loss unconditional of $\boldsymbol{x}_{\text{te}}$. We will start to generally define the squared bias

$$\mathbb{E}\left[(\boldsymbol{x}_{\text{te}}^{\text{T}}\boldsymbol{\beta}^g - \boldsymbol{x}_{\text{te}}^{\text{T}}(p\boldsymbol{\beta}^1 + (1-p)\boldsymbol{\beta}^0))^2\right] = \mathbb{E}[(\boldsymbol{x}_{\text{te}}^{\text{T}}\boldsymbol{\beta}_d^g)^2],$$

where $\boldsymbol{\beta}_d^g$ is the difference between the coefficients in the data-generating process of group $g$ and estimated coefficients. Note that in general

$$(\boldsymbol{x}_{\text{te}}^{\text{T}}\boldsymbol{\beta}_d^g)^2 = (\sum_{j=1}^p x_j \beta_{d,j}^g)(\sum_{j=1}^p x_j \beta_{d,j}^g) = \text{Tr}(\boldsymbol{x}_{\text{te}}\boldsymbol{x}_{\text{te}}^{\text{T}}\boldsymbol{\beta}_d^g \boldsymbol{\beta}_d^{g^{\text{T}}}),$$

$$\mathbb{E}[(\boldsymbol{x}_{\text{te}}^{\text{T}}\boldsymbol{\beta}_d^g)^2] = \text{Tr}(\mathbb{E}[\boldsymbol{x}_{\text{te}}\boldsymbol{x}_{\text{te}}^{\text{T}}]\mathbb{E}[\boldsymbol{\beta}_d^g \boldsymbol{\beta}_d^{g^{\text{T}}}]), \qquad \text{(Since they are independent)}$$

$$\mathbb{E}[\boldsymbol{x}_{\text{te}}\boldsymbol{x}_{\text{te}}^{\text{T}}] = \begin{pmatrix} 1 & \boldsymbol{0}^{\text{T}} \\ \boldsymbol{0} & \boldsymbol{\Sigma}. \end{pmatrix}.$$

Now, the challenge is to define $\boldsymbol{\beta}_d^g$. To do so, we define the difference between the coefficients as $\boldsymbol{\beta}_d = (\boldsymbol{\beta}^1 - \boldsymbol{\beta}^0)$. We can then define

$$(p\boldsymbol{\beta}^1 + (1-p)\boldsymbol{\beta}^0) = p(\boldsymbol{\beta}^1 - \boldsymbol{\beta}^0) + \boldsymbol{\beta}^0,$$

$$\boldsymbol{\beta}_d^{(1)} = \boldsymbol{\beta}^1 - p(\boldsymbol{\beta}^1 - \boldsymbol{\beta}^0) - \boldsymbol{\beta}^0 = (1-p)(\boldsymbol{\beta}^1 - \boldsymbol{\beta}^0),$$

$$\boldsymbol{\beta}_d^{(0)} = \boldsymbol{\beta}^0 - p(\boldsymbol{\beta}^1 - \boldsymbol{\beta}^0) - \boldsymbol{\beta}^0 = -p(\boldsymbol{\beta}^1 - \boldsymbol{\beta}^0),$$

$$\boldsymbol{\beta}_d^{(1)}\boldsymbol{\beta}_d^{(1)^{\text{T}}} = (1-p)^2 \boldsymbol{\beta}_d \boldsymbol{\beta}_d^{\text{T}}$$

$$\boldsymbol{\beta}_d^{(0)}\boldsymbol{\beta}_d^{(0)^{\text{T}}} = p^2 \boldsymbol{\beta}_d \boldsymbol{\beta}_d^{\text{T}}.$$

Because the difference in the coefficients is only in the intercepts, we can write

$$\mathbb{E}[(\boldsymbol{x}_{\text{te}}^{\text{T}}\boldsymbol{\beta}_d^{(1)})^2] = (1-p)^2 (a_1 - a_0)^2,$$

$$\mathbb{E}[(\boldsymbol{x}_{\text{te}}^{\text{T}}\boldsymbol{\beta}_d^{(0)})^2] = p^2 (a_1 - a_0)^2.$$

Suppose that we define $\boldsymbol{\Sigma}^{-1} = \frac{1}{\gamma}\boldsymbol{I}$. Note our DGP as defined in Section 4 uses $\gamma = 1$. We can now write

$$\mathbb{E}[\boldsymbol{x}_{\text{te}}^{\text{T}} \frac{\sigma^2}{n}(\frac{p^2}{p_{\text{tr}}} + \frac{(1-p)^2}{(1-p_{\text{tr}})})\boldsymbol{\Sigma}^{-1}\boldsymbol{x}_{\text{te}}] = \sigma^2 \left(\frac{p^2}{p_{\text{tr}}} + \frac{(1-p)^2}{(1-p_{\text{tr}})}\right) \frac{1}{n}\mathbb{E}[\boldsymbol{x}_{\text{te}}^{\text{T}} \begin{pmatrix} 1 & \boldsymbol{0}^{\text{T}} \\ \boldsymbol{0} & \frac{1}{\gamma}\boldsymbol{I} \end{pmatrix} \boldsymbol{x}_{\text{te}}]$$

$$= \sigma^2 \left(\frac{p^2}{p_{\text{tr}}} + \frac{(1-p)^2}{(1-p_{\text{tr}})}\right) \frac{d+1}{n}.$$

Now that we have defined the bias and variance terms, we can combine these with the noise term $\sigma^2$ into the expected loss.

$$\mathcal{B}^2(p_{\text{te}}, p, a_1, a_0) = p_{\text{te}}(1-p)^2(a_1 - a_0)^2 + (1-p_{\text{te}})p^2(a_1 - a_0)^2,$$

$$\mathcal{V}(\sigma^2, p, p_{\text{tr}}, n, d) = \sigma^2(\frac{p^2}{p_{\text{tr}}} + \frac{(1-p)^2}{(1-p_{\text{tr}})})\frac{d+1}{n},$$

$$\mathbb{E}_{D_{\text{tr}}^n \sim \mathcal{P}_{\text{tr}}}\left[\mathbb{E}_{(y,\boldsymbol{x})\sim\mathcal{P}_{\text{te}}}[(y - \boldsymbol{x}_{\text{te}}^{\text{T}}\hat{\boldsymbol{\beta}}(\boldsymbol{w}))^2 | D_{\text{tr}}]\right] \approx \mathcal{B}^2(p_{\text{te}}, p, a_1, a_0) + \mathcal{V}(\sigma^2, p, p_{\text{tr}}, n, d) + \sigma^2.$$

$$(37)$$

We then take the derivative of $p$ with respect to the bias,

$$\frac{\partial \mathcal{B}^2(p_{\text{te}}, p, a_1, a_0)}{\partial p} = 2(a_1 - a_0)^2(p - p_{\text{te}}),$$

$$2(a_1 - a_0)^2(p - p_{\text{te}}) = 0 \implies p = p_{\text{te}}.$$

In the case that $a_1 \neq a_0$, the bias is uniquely minimized when $p = p_{\text{te}}$. If we take the derivative of $p$ with respect to the variance, it is uniquely minimized at $p = p_{\text{tr}}$ if $n(p_{\text{tr}} - 1)p_{\text{tr}} \neq 0$.

$$\frac{\partial \mathcal{V}(\sigma^2, p, p_{\text{tr}}, n, d)}{\partial p} = -\frac{2(d+1)\sigma^2(p - p_{\text{tr}})}{n(p_{\text{tr}} - 1)p_{\text{tr}}}, \tag{38}$$

$$n(p_{\text{tr}} - 1)p_{\text{tr}} \neq 0, \frac{2(d+1)\sigma^2(p - p_{\text{tr}})}{n(p_{\text{tr}} - 1)p_{\text{tr}}} = 0 \implies p = p_{\text{tr}}. \tag{39}$$

For the variance, if $n \to \infty$, $\mathcal{V}(\sigma^2, p, p_{\text{tr}}, n, d) \to 0$. Hence, if our sample is large enough, the optimal $p$ will be $p$ for which the bias is minimal. The $p_n^*$ as per Equation 9 is derived by taking the derivative of the approximate expected loss with respect to $p$, and setting it to 0

$$n(p_{\text{tr}} - 1)p_{\text{tr}} \neq 0, \quad 2(a_1 - a_0)^2(p - p_{\text{te}}) - \frac{2(d+1)\sigma^2(p - p_{\text{tr}})}{n(p_{\text{tr}} - 1)p_{\text{tr}}} = 0, \tag{40}$$

$$\implies p = \frac{p_{\text{te}} + \eta\, p_{\text{tr}}}{1 + \eta}, \quad \text{where } \eta = \frac{\sigma^2(d+1)}{n(a_1 - a_0)^2 p_{\text{tr}}(1 - p_{\text{tr}})}. \tag{41}$$

### B.4 NUMERICAL COMPARISON

To illustrate the validity of the approximation used in the previous section, we compare it to the actual mean squared error via simulation. We simulate the mean squared error for the given DGP for the following values: $a_1 = 1, a_0 = 0, \sigma^2 = 1, \gamma = 1, p_{\text{tr}} = 0.9, p_{\text{te}} = 0.5$. We compare this to our approximation of the expected loss for several values of $n, d$. In addition, we plot the bias and variance. For 1,000 simulations, and several values of $n, d$, the results are shown in Figure 5.

We can observe in this figure that as $n$ decreases or $d$ increases, the variance becomes a greater share of the expected loss. This variance can be reduced by selecting a $p$ closer to the original $p_{\text{tr}}$. However, this subsequently leads to an increase in the bias. The optimal $p$ provides a trade-off between these two objectives.

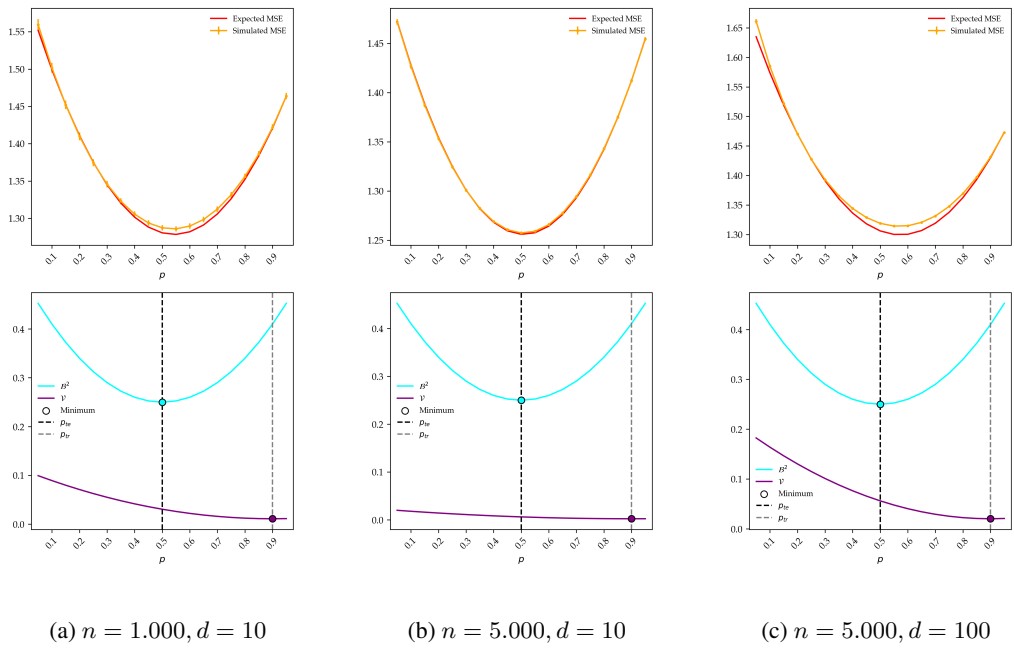

(a) $n = 1.000, d = 10$      (b) $n = 5.000, d = 10$      (c) $n = 5.000, d = 100$

Figure 5: Illustration of the bias-variance trade-off for $p_{\text{tr}} = 0.9, p_{\text{te}} = 0.5, a_1 = 1, a_0 = 0, \sigma^2 = 1, \gamma = 1$, and different values of $n$ and $d$. The simulated MSE is averaged over 1,000 runs. Error bars reflect the 95% Confidence interval.

## C  FULL SET OF RESULTS

| Method | Weighted Average Accuracy (%) | Worst Group Accuracy (%) |
|---|---|---|
| GW-ERM | 90.863 (0.351) | 82.556 (1.196) |
| +Optimized weights | 91.998 (0.092) | 85.435 (0.507) |
| Difference | 1.134 (0.287)** | 2.879 (0.878)** |
| SUBG | 90.892 (0.382) | 81.908 (1.503) |
| +Optimized weights | 91.709 (0.167) | 85.497 (0.543) |
| Difference | 0.817 (0.269)** | 3.589 (1.268)** |
| DFR | 90.629 (0.207) | 82.333 (0.735) |
| +Optimized weights | 91.039 (0.225) | 84.028 (0.978) |
| Difference | 0.410 (0.212)* | 1.695 (0.706)** |
| GDRO | 91.523 (0.188) | 84.306 (0.445) |
| +Optimized weights | 92.595 (0.139) | 88.481 (0.551) |
| Difference | 1.072 (0.258)** | 4.175 (0.393)** |
| JTT | 88.542 (0.326) | 79.319 (0.566) |
| +Optimized weights | 91.408 (0.396) | 84.925 (2.202) |
| Difference | 2.866 (0.466)** | 5.606 (1.986)** |

Table 1: Results for Waterbirds dataset of comparison between optimized and standard choice of weights: we report the average over 5 runs, with the standard error in parentheses. The */** indicate statistical significance at a 10%/5% significance level respectively for a paired sample one-sided $t-$test.

| Method | Weighted Average Accuracy (%) | Worst Group Accuracy (%) |
|---|---|---|
| GW-ERM | 90.471 (0.120) | 85.667 (0.539) |
| +Optimized weights | 90.700 (0.062) | 87.000 (0.136) |
| Difference | 0.229 (0.114)* | 1.333 (0.487)** |
| SUBG | 89.141 (0.111) | 87.598 (0.284) |
| +Optimized weights | 89.238 (0.162) | 83.333 (0.731) |
| Difference | 0.097 (0.185) | -4.265 (0.734) |
| DFR | 88.320 (0.195) | 81.000 (0.920) |
| +Optimized weights | 88.299 (0.176) | 81.171 (1.698) |
| Difference | -0.021 (0.092) | 0.171 (1.118) |
| GDRO | 89.113 (0.173) | 85.333 (0.648) |
| +Optimized weights | 90.712 (0.083) | 89.667 (0.222) |
| Difference | 1.599 (0.203)** | 4.333 (0.727)** |
| JTT | 88.496 (0.121) | 79.222 (0.716) |
| +Optimized weights | 88.243 (0.165) | 75.556 (0.680) |
| Difference | -0.253 (0.139) | -3.667 (0.372) |

Table 2: Results for CelebA dataset of comparison between optimized and standard choice of weights: we report the average over 5 runs, with the standard error in parentheses. The */** indicate statistical significance at a 10%/5% significance level respectively for a paired sample one-sided $t-$test.

| Method | Weighted Average Accuracy (%) | Worst Group Accuracy (%) |
|---|---|---|
| GW-ERM | 83.967 (0.192) | 72.087 (0.779) |
| +Optimized weights | 84.315 (0.238) | 76.114 (0.660) |
| Difference | 0.349 (0.106)** | 4.027 (0.185)** |
| SUBG | 84.100 (0.233) | 73.058 (1.085) |
| +Optimized weights | 84.287 (0.169) | 74.163 (0.798) |
| Difference | 0.187 (0.092)* | 1.105 (0.621)* |
| DFR | 84.281 (0.109) | 78.207 (0.313) |
| +Optimized weights | 84.654 (0.216) | 79.740 (0.130) |
| Difference | 0.373 (0.162)** | 1.533 (0.367)** |
| GDRO | 84.452 (0.208) | 75.654 (0.585) |
| +Optimized weights | 84.400 (0.190) | 76.350 (0.472) |
| Difference | -0.051 (0.085) | 0.696 (0.338)* |
| JTT | 84.071 (0.205) | 72.427 (0.680) |
| +Optimized weights | 84.374 (0.257) | 76.226 (0.698) |
| Difference | 0.303 (0.081)** | 3.799 (0.211)** |

Table 3: Results for MultiNLI dataset of comparison between optimized and standard choice of weights: we report the average over 5 runs, with the standard error in parentheses. The $*/**$ indicate statistical significance at a 10%/5% significance level respectively for a paired sample one-sided $t-$test.

# D    OPTIMIZING THE WEIGHTS FOR THE WORST-GROUP LOSS

Here we present the algorithm for GDRO with optimized weights. It should be read in conjunction with Section 5.3.

---

**Algorithm 2:** Estimation of the optimal weights $\hat{\boldsymbol{p}}_n^*$ for GDRO

---

**Input:** data $\{y_i, g_i, \boldsymbol{x}_i\}_{i=1}^n$, training set size $n' < n$, maximum steps $T$, learning rates $\eta_{\boldsymbol{p}}, \eta_{\boldsymbol{q}}$ and momentum $\gamma$.

Initialize $\boldsymbol{p}_0$, $\boldsymbol{q}_0$ uniformly and normalized to 1.

Split the data into a training set of size $n'$ and a validation set.

**for** $t = 1, \ldots, T$ **do**

    Estimate $\hat{\boldsymbol{\theta}}_{n'}(\boldsymbol{p}_{t-1})$ as in Equation 1.

    Compute hyper-gradient $\boldsymbol{\zeta}_t = -\nabla_{\boldsymbol{p}_{t-1}} \mathcal{L}_{\text{val}}(\hat{\boldsymbol{\theta}}_{n'}(\boldsymbol{p}_{t-1}), \boldsymbol{q}_{t-1})$ with implicit function theorem.

    Update weights $p_{t,g} = p_{t-1,g} \exp(\eta u_{t,g})$, with $u_{t,g} = \gamma u_{t-1,g} + (1 - \gamma)\zeta_{t,g}$, for all $g \in \mathcal{G}$.

    Normalize $p_{t,g} = \frac{p_{t,g}}{\sum_{g' \in \mathcal{G}} p_{t,g'}}$ for all $g \in \mathcal{G}$.

    Update loss weights $q_{t,g} = q_{t-1,g} \exp\left(\eta_{\boldsymbol{q}} \mathcal{L}_g(\hat{\boldsymbol{\theta}}_{n'}(\boldsymbol{p}_{t-1}))\right)$ for all $g \in \mathcal{G}$, where

    $\mathcal{L}_g(\boldsymbol{\theta}) := \frac{1}{n_g} \sum_{i=1}^{n_g} \mathcal{L}(y_i^g, f_{\boldsymbol{\theta}}(\boldsymbol{x}_i^g))$ and with $n_g, y_i^g$ and $\boldsymbol{x}_i^g$ as in Section 5.2.

    Normalize $q_{t,g} = \frac{q_{t,g}}{\sum_{g' \in \mathcal{G}} q_{t,g'}}$ for all $g \in \mathcal{G}$.

**end**

Return $\hat{\boldsymbol{p}}_n^* = \arg\min_{\boldsymbol{p}_t \in \{\boldsymbol{p}_0, \boldsymbol{p}_1, \ldots, \boldsymbol{p}_T\}} \max_{g \in \mathcal{G}} \mathcal{L}_g(\hat{\boldsymbol{\theta}}_{n'}(\boldsymbol{p}_t))$.

---

# E    EXPERIMENTAL DETAILS

## E.1    DATASETS

An overview of the datasets, their group definitions and sizes, as well as the train/validation split used in the experiments, is provided in Table 4.

|  | Target variable | Counts (training set) | | Percentage | | Dataset size | |
|---|---|---|---|---|---|---|---|
| Waterbirds |  | Water | Land |  |  | Train | Val |
|  | Water bird | 1,057 | 56 | 22.04% | 1.17% | 4,795 | 1,199 |
|  | Land bird | 184 | 3,498 | 3.84% | 72.95% |  |  |
| CelebA |  | Female | Male |  |  | Train | Val |
|  | Blonde | 22,880 | 1,387 | 14.06% | 0.85% | 162,770 | 19,867 |
|  | Not blonde | 71,629 | 66,874 | 44.01% | 41.08% |  |  |
| MultiNLI |  | No negation | Negation |  |  | Train | Val |
|  | Entailment + Neutral | 24,362 | 638 | 48.72% | 1.28% | 50,000 | 20,000 |
|  | Contradiction | 20,930 | 4,070 | 41.86% | 8.14% |  |  |

Table 4: Overview of all datasets and their respective group sizes.

**Waterbirds**: this dataset from Sagawa et al. (2020a) is a combination of the Places dataset (Zhou et al., 2016) and the CUB dataset (Welinder et al., 2010). A 'water background' is set by selecting an image from the lake and ocean categories in the places dataset, and the 'land background' is set based on the broadleaf and bamboo forest categories. A waterbird/landbird is then pasted in front of the background. The goal is to predict whether or not a picture contains a waterbird or landbird.

As explained in Section 6, we deviate from Sagawa et al. (2020a) in that our validation set is not balanced for all groups. This setup of Sagawa et al. (2020a) gives an advantage to methods that only fit on the validation set, such as DFR.

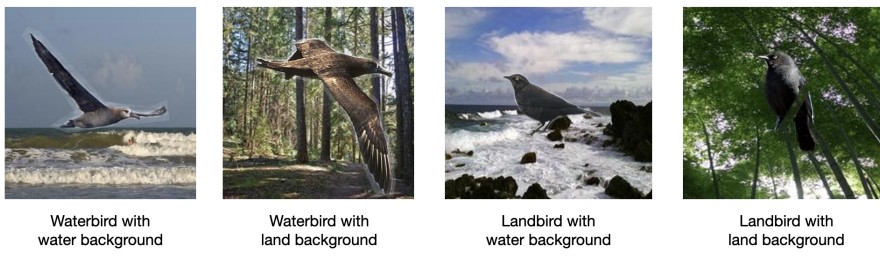

Figure 6: Examples of the different images in the Waterbirds dataset.

**CelebA**: this dataset contains images of celebrity faces (Liu et al., 2015). The goal is to predict whether or not a picture contains a celebrity with blonde or non-blonde hair. This is spuriously correlated with the sex of the celebrity.

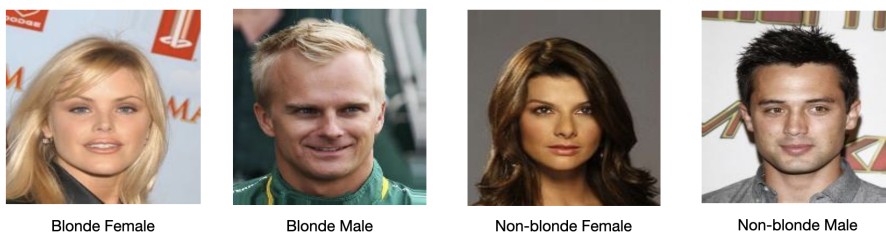

Figure 7: Examples of the different images in the CelebA dataset.

**MultiNLI**: the MultiNLI dataset (Williams et al., 2018) contains pairs of hypotheses and premises. The sentences are pasted together with a [SEP] token in between. The original dataset contains three label types: negation, entailment, and neutral. We follow a procedure that is similar to Kumar et al. (2022) and Holstege et al. (2024) and create a smaller binary version of the dataset (50.000 samples in training, 20.000 in validation, 30.000 in test), with only negations ($y = 0$) or entailment or neutral ($y = 1$). This version of the dataset is class balanced, similar to the original MultiNLI dataset. It has been reported that the negation label is spuriously correlated with negation words like nobody, no, never and nothing (Gururangan et al., 2018). A binary label is created to denote the presence of these words in the hypothesis of a given hypothesis-premise pair.

### E.2 TRAINING OF DNNS

**Models**: For the Waterbirds and CelebA dataset, we use the Resnet50 architecture implemented in the `torchvision` package: `torchvision.models.resnet50(pretrained=True)`. More details on the model can be found in the original paper from He et al. (2016). For Waterbirds, this means using a learning rate of $10^{-3}$, a weight decay of $10^{-3}$, a batch size of 32, and for 100 epochs without early stopping. For CelebA, this means using a learning rate of $10^{-3}$, a weight decay of $10^{-4}$, a batch size of 128, and for 50 epochs without early stopping. We use stochastic gradient descent (SGD) with a momentum parameter of 0.9. For simplicity, we perform no data augmentation for both datasets.

For the MultiNLI dataset, we use the base BERT model implemented in the `transformers` package (Wolf et al., 2019): `BertModel.from_pretrained("bert-base-uncased")`. The model was pre-trained on BookCorpus, a dataset consisting of 11,038 unpublished books, as well as the English Wikipedia (excluding lists, tables and headers). More details on the model can be found in the original paper: Devlin et al. (2019a). For finetuning the BERT model on MultiNLI, we use the AdamW optimizer (Loshchilov & Hutter, 2017) with the standard settings in *Pytorch*. When finetuning, we use the hyperparameters of Izmailov et al. (2022), training for 10 epochs with a batch size of 16, a learning rate of $10^{-5}$, and a weight decay of $10^{-4}$, and linear learning rate decay.

### E.3 IMPLEMENTATION DETAILS OF EXISTING METHODS

For all methods, unless otherwise mentioned, we use a logistic regression with the `sklearn.LogisticRegression` class with the `Liblinear` solver to optimize the logistic regression, with a tolerance of $10^{-4}$.

In each case, we demean the training data and scale it such that each column has a variance of 1. We use the estimated mean and variances of the training data to apply the same transformations to the validation and test data.

For each run, the hyperparameters are selected based on the worst-group accuracy on the respective validation set. For the L1 penalty, we select it from the following values: $0.1, 1.0, 3.3, 10.0, 33.33, 100, 300, 500$. In the case that multiple L1 penalties lead to the same worst-group accuracy, we select the highest one.

**GW-ERM and SUBG**: we only optimize the L1 penalty.

**GDRO**: We follow the implementation of Sagawa et al. (2020a). We optimize two hyperparameters: the L1 penalty, and the hyperparameter $C$ used in adapted worst-group loss from Sagawa et al. (2020a), with the possible values of $C$ being $0, 1, 2, 3$. Instead of the `Liblinear` solver, we use gradient descent with a learning rate of $10^{-5}$. After observing that the standard options for the L1 penalty often led to poor performance, we add the values $0.01, 0.05, 0.2$ to the possible L1 penalties.

**DFR**: We follow the implementation of Kirichenko et al. (2023). We use the validation set of each dataset for fitting the model. For hyperparameter selection, we split this validation set in half, and use one half for selecting the L1 penalty. After this, the entire validation set is used to fit the model. Parameters are estimated by averaging over 10 logistic regression models.

**JTT**: similar to Liu et al. (2021) we train a DNN, of which the errors are used to identify the groups. For Waterbirds and CelebA, we train a Resnet50 model using the hyperparameters chosen by Liu et al. (2021). In the case of Waterbirds, this means a learning rate of $10^{-5}$ and a weight decay of 1. In the case of CelebA this means a learning rate of $10^{-5}$ and a weight decay of 0.1. In the case of

MultiNLI, we train a BERT model with the same hyperparameters that were used for finetuning. In all cases, we use early stopping.

After getting the errors from the respective model, we define the groups accordingly and retrain the last layer on the embeddings that are used for all other methods. This is in order to ensure a fair comparison between methods. We optimize over both the L1 penalty and the upweighting factor $\lambda_{JTT}$. For the latter hyperparameter, we select from the values $1, 2, 5, 10, 25$.

### E.4 IMPLEMENTATION DETAILS OF OPTIMIZING THE WEIGHTS

Following the notation in Algorithm 1, we use a learning rate $\eta$ of 0.1 for each dataset and method, except in two cases: optimizing the weights for DFR on Waterbirds and MultiNLI, where we use a learning rate of 0.01 and 0.2 respectively. This was done after observing that the optimization procedure did not converge to a lower validation loss for the original choice of the learning rate. In the case of Waterbirds, we run Algorithm 1 for $T = 200$ steps, and for $T = 100$ in the case of CelebA and MultiNLI.

For each dataset and method, we use a momentum parameter of $\gamma = 0.5$, after observing this helped with escaping local minima.

In the case of optimizing the worst-group loss, we use $\eta_q = 0.1$, following Sagawa et al. (2020a), and use Algorithm 2 while using GW-ERM as our method for determining the parameters.

In the case of optimizing the weights of SUBG and DFR, we set $v_g = 1$ for the group $g$ with the smallest number of samples. We then optimize the other fractions of groups.

