# OpenReview forum: "Optimizing importance weighting in the presence of sub-population shifts"
_ICLR.cc/2025/Conference — ICLR 2025 Poster_

### Official Review · Reviewer_PJmP · 2024-10-28

**Soundness:** 4
**Presentation:** 4
**Contribution:** 3
**Rating:** 8
**Confidence:** 2

**Summary:**

The authors present a bi-level optimization strategy to estimate optimal importance weights in the presence of distribution shifts. The authors use a linear regression model to demonstrate how the naïve but often used ratio of likelihoods is not necessarily an optimal choice for importance weights (figure 1.) The authors proceed to introduce their bi-level optimization strategy that estimates optimal weights given a validation set sampled from the training set.

The authors emphasize that the strategy can be used in conjunction with existing methods and write how they implement their strategy specific to each of five existing methods for fine-tuning models to manage distribution shift. The authors empirically test their claim using a benchmark of three datasets for each of the 5 modified methods and show how in almost all cases the performance on a held-out test dataset is improved using their bi-level optimization strategy.

**Strengths:**

To me the paper is a clear acceptance: it is well written, organized, to the point. You present an intriguing argument that the naïve choice of importance weights used in a variety of prior algorithms can be improved upon, present a practical answer of how to define and find these optimal weights, then show that this definition brings benefit over a controlled baseline. The topic clearly has potential for high impact as distribution shifts are pretty ubiquitous.

**Weaknesses:**

I think the paper could be improved with some additional discussion about the practical implementation of these strategies. For example, how much additional time is spent on your specific improvements compared to the naïve/heruristic weights? Roughly how does this time scale with the validation dataset fraction (1 – n’/n)? What additional hyperparameter tuning is done to find optimal weights (Appendix E.4)? Would a user need to repeat this hyperparameter tuning for their own model?

The impact of the paper could be heightened if the authors elected to demonstrate their technique improving upon a real-world challenge; for example, the authors identify in their introduction that importance weighting technique could be used to improve outcomes for algorithmic fairness.

**Questions:**

I’ll focus specifically on the empirical experiments. I deem them overall sound, but I have a few questions to be sure. The authors describe fine-tuning a network 5 times to build a sample and perform their t-test. I wanted to confirm how that network was initialized (Appendix suggests you had the same base network that you then fine-tuned 5 different times?). It was then from this set of 5 fine-tuned networks that you then retrained the last layer using each algorithm’s original importance weight scheme and then compared to the paired performance when replacing with your optimal weights? What was controlled in this experiment in terms of random seeds? Were seeds controlling batch ordering varied between runs? Were seeds being varied between the pairs of optimal vs original weights?

The appendix describes some hyperparameters that had to be found for the optimal weight algorithm (E.4). Can the authors provide some additional detail about how these hyper parameters are found?

---

> ### Author Response · Authors · 2024-11-15
>
> We are deeply grateful to reviewer PJmP for the detailed feedback and questions. Below, we address these point-by-point.
>
> >How much additional time is spent on your specific improvements compared to the naïve/heruristic weights? Roughly how does this time scale with the validation dataset fraction (1 – n’/n)?
>
> We thank the reviewer for raising this question. We have discussed the computational overhead of our optimization procedure in response to reviewer Ng11. Generally speaking, provided we restrict ourselves to last-layer retraining, the computational overhead of our optimization procedure is manageable. To address the specific question of reviewer PJmP, increasing $n'$ relative to $n_{\mathrm{val}}$ (and thus $n$) will increase the computational overhead. This is because $n_{\mathrm{val}}$ plays no role in re-fitting the logistic regression, which is the most time-consuming part of Algorithm 1. Moreover, $n_{\mathrm{val}}$ plays no role in calculating and inverting the Hessian.

---

> ### Author Response · Authors · 2024-11-15
>
> >What additional hyperparameter tuning is done to find optimal weights (Appendix E.4)? Would a user need to repeat this hyperparameter tuning for their own model?
>
> >Can the authors provide some additional detail about how these hyper parameters are found?
>
> In order to select the the learning rate $\eta$ and the momentum parameter $\gamma$, we observed how the validation loss progressed over the course of the optimization for given iterations $t=1,2,\ldots,T.$ We observed that for some choices of learning rates $\eta$, the validation loss would barely decrease over the course of the optimization. We also observed that if we used no momentum, the optimization sometimes would halt around a particular choice of weights. In all but two cases (optimizing the weights for DFR on Waterbirds and MultiNLI) we observed that for the relatively standard choices of $\eta=0.1$ and $\gamma=0.5$, the validation loss decreased throughout the optimization training procedure, until it plateaued towards the end of the optimization. For the two exceptions, we changed the learning rate to a value (0.01 and 0.2, respectively) for which we observed a similar phenomenon of convergence.
>
> We acknowledge that we did not pursue the most rigorous approach for selecting the additional hyperparameters, although we did follow the standard approach of using the validation set to assess the choices of the hyperparameters. More rigorous would be to conduct a sweep over different values of $\eta$ and $\gamma,$ and observe for which combination of these values the validation loss is lowest. The risk of this approach is that one might 'overfit' on the validation set. This issue can be remedied by existing approaches for preventing overfitting, such as early stopping.
>
> *(see next comment for a continuation of this discussion)*

---

> ### Author Response · Authors · 2024-11-15
>
> *(continuation of previous discussion)*
>
> To show that this approach for selecting the learning rate and momentum can work, we have conducted an additional experiment for the Waterbirds dataset, where we optimize the weights for group-weighted empirical risk minimization (GW-ERM). We keep the setup (i.e., averaging over 5 seeds) and all other hyper-parameters the same (e.g. the $l_1$ penalty strength and $T=200$) and only vary the values of the learning rate and momentum. We show the results in the table below. The bottom row of the table indicates the performance if one selects per seed the best combination of the learning rate and momentum, based on the weighted validation loss.
>
> The table below indicates that the results are relatively robust to the choice of the hyper-parameters. For instance, even the worst combination of learning rate and momentum would result in an improvement (on average) compared to the standard weights. Moreover, this worst combination could be avoided by the user by selecting the learning rate and momentum based on the weighted validation loss. If one were to follow this strategy, this would even lead to an improvement compared to our choice of hyperparameters and the results presented in Figure 2 of the manuscript. We will report this table in Appendix E.4 of an updated version of the manuscript.
>
> | Learning Rate $\eta$                        | Momentum $\gamma$ | Weighted Validation Loss | Weighted Test Loss | Weighted Average Test Accuracy | Worst-Group Test Accuracy |
> | -------------------------------------------- | ------------------ | ------------------------ | ------------------ | ------------------------------ | ------------------------- |
> | 0.01                                         | 0.0                | 0.224 (0.014)            | 0.259 (0.018)      | 91.100 (0.313)                 | 82.991 (0.873)            |
> | 0.01                                         | 0.5                | 0.226 (0.013)            | 0.261 (0.018)      | 91.141 (0.310)                 | 82.928 (0.924)            |
> | 0.01                                         | 0.9                | 0.225 (0.013)            | 0.260 (0.019)      | 91.102 (0.322)                 | 83.084 (0.982)            |
> | 0.10                                         | 0.0                | 0.206 (0.016)            | 0.223 (0.010)      | 92.173 (0.165)                 | 86.160 (0.396)            |
> | 0.10                                         | 0.5                | 0.205 (0.016)            | 0.229 (0.010)      | 91.998 (0.092)                 | 85.435 (0.507)            |
> | 0.10                                         | 0.9                | 0.206 (0.015)            | 0.222 (0.011)      | 92.211 (0.201)                 | 86.494 (1.050)            |
> | 0.20                                         | 0.0                | 0.204 (0.016)            | 0.217 (0.012)      | 92.479 (0.139)                 | 87.431 (0.811)            |
> | 0.20                                         | 0.5                | 0.205 (0.016)            | 0.219 (0.011)      | 92.326 (0.191)                 | 86.698 (0.777)            |
> | 0.20                                         | 0.9                | 0.205 (0.016)            | 0.217 (0.012)      | 92.430 (0.158)                 | 87.477 (1.055)            |
> | Using best weighted validation loss per seed |                    | 0.204 (0.016)            | 0.222 (0.011)     | 92.144 (0.163)                 | 86.339 (0.757)
>
> With regards to the number of steps $T$, we have been pragmatic. We chose a $T$ large enough such that we observed convergence of the validation loss, but not too large for our computational budget and time available.

---

> ### Author Response · Authors · 2024-11-15
>
> >The impact of the paper could be heightened if the authors elected to demonstrate their technique improving upon a real-world challenge; for example, the authors identify in their introduction that importance weighting technique could be used to improve outcomes for algorithmic fairness.
>
> We welcome this suggestion by the reviewer, and hope to address it by adding results of using our procedure on a dataset in the medical domain, related to algorithmic fairness. In an updated version of the manuscript, we will also add a sentence to the introduction to highlight how importance weighting can play a role in achieving algorithmic fairness.
>
> >The authors describe fine-tuning a network 5 times to build a sample and perform their t-test. I wanted to confirm how that network was initialized...What was controlled in this experiment in terms of random seeds? Were seeds controlling batch ordering varied between runs? Were seeds being varied between the pairs of optimal vs original weights
>
> For our vision experiments we used Resnet50, with parameters initialized from a model that is pre-trained on Imagenet1K (downloaded from the torchvision package). In the case of the BERT model (used for multiNLI), the parameters are initialized from a model that was pre-trained on BookCorpus, a dataset consisting of 11,038 unpublished books, as well as the English Wikipedia (excluding lists, tables and headers). So, for each of the seeds, the initial parameters of the deep neural network (DNN) are the same.
>
> The seed was set before the finetuning of the DNN started. This means that per seed, the ordering of batches and which samples are contained in them changes. After the DNN was finetuned, we stored the embeddings of the last layer for each seed. Then, a logistic regression is trained on these embeddings - both with standard, and optimized weights. Thus, the seeds are **not** varied between pairs of standard and optimal weights. In this sense, we performed a paired t-test.
>
> We will clarify both how the DNNs were initialized and which parts of the training run were controlled with random seeds in Appendix E.2 of the updated manuscript.

---

### Official Review · Reviewer_DCrk · 2024-11-03

**Soundness:** 2
**Presentation:** 2
**Contribution:** 2
**Rating:** 6
**Confidence:** 3

**Summary:**

This work studies importance weighting of sub-population shifts. Conventional naive importance reweighting is $p_{te}(x,y)/ p_{tr}(x,y)$. This work argues that this conventional reweighting is sub-optimal under a limited training data size. Thus this work proposes to uses a learned importance reweighting coefficient. Empirical results show improved generalization performance across benchmarks like image and NLP datasets.

**Strengths:**

- The arguement that "the conventional importance weighting $p_{te}(x,y)/ p_{tr}(x,y)$ could be sub-optimal" is interesting.

- This paper estimates model parameters and importance weights iteratively on train and iid-validation datasets. That is interesting, because it reduces the overfitting of training dataset. Different from conventional cross-validation and hyper-parameter search, this approach find the importance weights by optimization.

**Weaknesses:**

- Experiments shows the good performance of the proposed method. However, Algorithm 1 (line 278 $p_0$ and line 282 $r$) requires the access of test dataset as parameter initialization. This could be problematic. I understand that some explicit reweighting methods, such as JTT, also contains some heuristic & data-dependent importance weights searching space. These heuristic hyperparameter searching space could also depends on test dataset.  However, group distributional robust optimization (GDRO) links to importance weighting in an implicit way [1], and doesn't contain any hints from test dataset. Thus, the comparison between the proposed method and GDRO is prolematic and potentionally unfair.

- minor typos: line 112 [L(y, fθ (x)] --> [L(y, fθ (x))]


[1] Zhang, J., Lopez-Paz, D., & Bottou, L. (2022, June). Rich feature construction for the optimization-generalization dilemma. In International Conference on Machine Learning (pp. 26397-26411). PMLR.

**Questions:**

- How important are the $p_0$ and $r$ in Algorithm 1 line 278 and line 282? Currently these parameters depend on test dataset directly. And thus weaken the usefulness of the algorithm 1. Is it possible to set $p_0$ and $r$ by assuming a test dataset $p_{te}(g_i) = p_{te}(g_j) , \forall i,j $? If it is possible, how does it affect the experiments performances?  That is my main concern.

---

> ### Author Response · Authors · 2024-11-15
>
> We are thankful to reviewer DCrk for carefully reading the manuscript and their feedback. Below, we address the feedback point-by-point.
>
> >Experiments shows the good performance of the proposed method. However, Algorithm 1 (line 278 $p_0$ and line 282 $r$) requires the access of test dataset as parameter initialization. This could be problematic.
>
> >How important are the p0 and r in Algorithm 1 line 278 and line 282? Currently these parameters depend on test dataset directly. And thus weaken the usefulness of the algorithm 1. Is it possible to set p0 and r by assuming a test dataset pte(gi)=pte(gj),∀i,j? If it is possible, how does it affect the experiments performances? That is my main concern
>
> In the above two paragraphs, the reviewer is raising the issue that the user might not have access to data from the test distribution of interest, and thus might not know the probability at which certain groups occur ($p_{\mathrm{te}}(g)$). We thank the reviewer for pointing this out and we agree with the reviewer that this could be problematic. However, we do believe there are many cases in which it is possible to apply Algorithm 1.
>
> For the experiments in the manuscript we do exactly as the reviewer suggests and assume that the test distribution of interest is one where each group occurs to the same extent - i.e., $\forall g \in\mathcal{G},  p_{\mathrm{te}}(g) = \frac{1}{G}$, where $G = |\mathcal{G}|$. This is in line with previous work [1] that also uses these benchmarks, where performance of the methods is (partially) assessed by an equal-weight average of the accuracy per group. The motivation behind this is that if a model relies on a spurious correlation, e.g. using the water background to classify a waterbird type, it should perform badly on test dataset where is spurious correlation is absent, i.e. when all groups have equal marginal probabilities.
>
> In more practical contexts, the probability at which certain groups occur could be informed by knowledge of the user about the prediction problem. For example, the user knows that male patients are more prevalent than female patients. If a small subset of labels from the test distribution is available, a user could estimate the likelihood ratio (used for $p_0$ and $r$ in Algorithm 1) of each group occuring.
>
> And even if data from the test distribution is not available, a user can still specify the test distribution of interest. The test distribution then reflects the preferences of the user. For example, if a user cares to an equal extent about prediction errors in two groups (e.g., male and female patients), they can specify this via a test distribution in which the groups have an equal marginal probability.
>
> Another alternative would be to optimize the weights for worst-group loss, instead of the weighted average accuracy. This is the GDRO approach with optimized weights, as outlined in Section 5.3 of the paper and Algorithm 2 in Appendix D. In this case, no knowledge of the test distribution is required.
>
> Finally, we believe that the fact that the starting values of $p_0$ in Algorithms 1 and 2 are determined by the test distribution, as pointed out by the reviewer, is a logical choice. As outlined in Sections 3 and 4, this choice is optimal in the case of infinite sample size ($n \rightarrow \infty$). The bi-level optimization procedures (Algorithms 1 and 2) are thus initialized at the heuristic choice of the weights and converge to the optimal weights by deviating from this initial choice. We will clarify the motivation for this choice in an updated version of the manuscript.
>
> [1] Badr Youbi Idrissi, Martin Arjovsky, Mohammad Pezeshki, and David Lopez-Paz. Simple data balancing achieves competitive worst-group-accuracy. In Bernhard Scholkopf, Caroline Uhler and Kun Zhang (eds.), Proceedings of the First Conference on Causal Learning and Reasoning, volume 177 of Proceedings of Machine Learning Research, pp. 336–351. PMLR, 11–13 Apr 2022.

---

> ### Author Response · Authors · 2024-11-15
>
> >However, group distributional robust optimization (GDRO) links to importance weighting in an implicit way [2], and doesn't contain any hints from test dataset. Thus, the comparison between the proposed method and GDRO is prolematic and potentionally unfair.
>
> This point requires some clarification from our side. The application of our weight optimization procedure to GDRO is described in Section 5.3 and Algorithm 2 in Appendix D of the manuscript. In Algorithm 2 no knowledge of the test distribution is being used. Contrary to Algorithm 1, the likelihood ratio $r$ is not used in Algorithm 2. Instead, and in accordance with the standard GDRO approach, we optimize the weights $\boldsymbol{p}$ for worst-group loss (Equation 13 of the manuscript). For the initial value $p_0$, we simply use the same probability for each group, i.e., $\forall g \in \mathcal{G}, p_{g,0} = \frac{1}{|\mathcal{G}|}.$ This choice can be made regardless of the marginal probabilities of a (hypothetical) test distribution. We thus believe the comparison with GDRO is fair. We agree with the reviewer that in the manuscript this point is not explained with enough clarity and will rectify this in an updated version of the manuscript.
>
> >minor typos: line 112 [L(y, fθ (x)] --> [L(y, fθ (x))]
>
> We thank reviewer DCrk for pointing out the typo, and will rectify this in the updated manuscript.
>
> [2] Zhang, J., Lopez-Paz, D., & Bottou, L. (2022, June). Rich feature construction for the optimization-generalization dilemma. In International Conference on Machine Learning (pp. 26397-26411). PMLR.

---

> ### Comment · Reviewer_DCrk · 2024-11-26
>
> Thank authors for the response. The point that GDRO experiments use a uniform p initialization is convincing, even though estimating p from test set is problematic and not solved.
>
> It would be very interesting to initializate p uniformly on all tasks. but...
>
> Overall, I would like to raise my score, because GDRO experiment partically shows the effect of uniform initialization.

---

> > ### Author Response · Authors · 2024-11-27
> >
> > We thank reviewer DCrk for the constructive feedback and raising their score.
> >
> > We agree with the reviewer that “estimating p from test set” is a limitation of our procedure for optimizing weights. We do want to note, however, that apart from GDRO and JTT the standard importance weighting techniques discussed in the manuscript (GW-ERM, SUBG, DFR) also implicitly assume this knowledge about the test distribution. Based on the initial feedback of the reviewer, we have added a paragraph in the Conclusion and Discussion section (line 519) where we explicitly discuss the limitation and possible ways to deal with it. We thank the reviewer for pointing out this omission in the original version of the manuscript.
> >
> > The reviewer also suggests “to initialize p uniformly on all tasks”. We would like to reiterate that this is the setup of all experiments in the manuscript. To be more concrete, the initial p used in all experiments of the manuscript is given by $p_{0,g} = \frac{1}{G}$, where $G = |\mathcal{G}|,$ $\forall g \in\mathcal{G}$ (for the experiments based on Algorithm 1, this is because we use a test distribution of interest with uniform marginal probabilities).

---

### Official Review · Reviewer_xwkt · 2024-11-04

**Soundness:** 3
**Presentation:** 3
**Contribution:** 3
**Rating:** 8
**Confidence:** 2

**Summary:**

In this work, the authors develop a bi-level optimization algorithm for estimating importance weights (group weights) on top of existing deep learning algorithms, such as Group Weighted ERM and GDRO, to enhance generalization under sub-population shifts. They utilize a standard linear model to illustrate the necessity of this optimization in the presence of limited data and provide mathematical proofs to support their claims. Empirical demonstrations with deep networks using existing training strategies are presented, showing performance improvements on standard datasets.

**Strengths:**

The paper is well-written and systematically defines all terms and equations with attention to detail. It is mathematically well-supported and contains relevant experiments on standard benchmarks.

**Weaknesses:**

Refer to Questions

**Questions:**

* Can the authors clarify the overall outline of the training procedure? Did they follow the last-layer retraining protocol as described in Izamailov et al. (2022)? If so, additional details about this protocol would be beneficial.

* For the Group Weighted ERM procedure, it is mentioned that the likelihood ratio between the test and training distributions is required to initiate the proposed algorithm. However, in most cases, access to the test distribution is not available. How is this ratio computed for real-world datasets, such as Waterbirds? (I may have missed a key concept here.)

* Can the authors provide qualitative insights on how they believe the performance of their algorithms may vary with an increase in model scale, particularly for architectures like ViTs and SWIN?

* Can the authors include an example with large-scale datasets, such as ImageNet-1K, or a medical imaging benchmark (e.g., ISIC or MedMNIST) to enhance the impact of the proposed algorithms?

## Post-Rebuttal
I have increased my score and recommend acceptance

---

> ### Author Response · Authors · 2024-11-15
>
> We thank reviewer Xwkt for carefully reading the manuscript and their feedback. Below, we address the feedback point-by-point.
>
> >Can the authors clarify the overall outline of the training procedure? Did they follow the last-layer retraining protocol as described in Izmailov et al. (2022)?
>
> We indeed perform last-layer retraining in a manner similar to Izmailov et al. [1]. The steps in our training procedure are:
> 1. We start with a deep neural network (DNN) that is pre-trained on a large dataset (e.g. Imagenet1k).
> 2. We finetune the DNN on the respective datasets of our experiments.
> 3. We store the embeddings of the last layer of the DNN.
> 4. We then train a logistic regression on these embeddings, which is equivalent to retraining the last layer.
>
> With regards to the logistic regression of step 4, we apply all the respective importance weighting techniques (GW-ERM, SUBG, DFR, GDRO, JTT), using both standard and optimized weights. We will clarify this procedure by updating Appendix E.2 of the manuscript.
>
> >For the Group Weighted ERM procedure, it is mentioned that the likelihood ratio between the test and training distributions is required to initiate the proposed algorithm. However, in most cases, access to the test distribution is not available. How is this ratio computed for real-world datasets, such as Waterbirds?
>
> We thank the author for raising the issue of the choice of the test distribution, and how this related to the benchmark datasets.
>
> For all benchmark datasets, we assume that the test distribution of interest is one where each group has the same probability of occuring - i.e., $\forall g \in\mathcal{G},  p_{\mathrm{te}}(g) = \frac{1}{G}$, where $G = |\mathcal{G}|$. This is in line with previous work [1] that also uses these benchmarks, where performance of the methods is (partially) assessed by an equal-weight average of the accuracy per group. The motivation behind this is that if a model relies on a spurious correlation, e.g. using the water background to classify a waterbird type, it should perform badly on test dataset where is spurious correlation is absent, i.e. when all groups have equal marginal probabilities.
>
> In response to reviewer DCrk we have outlined some general considerations for how a user might select the test distribution of interest.
>
> >Can the authors provide qualitative insights on how they believe the performance of their algorithms may vary with an increase in model scale, particularly for architectures like ViTs and SWIN?
>
> Because we resort to last-layer retraining, only the phase of finetuning the DNN is affected by the scale of the model. If the last-layer embeddings of the larger model also have a larger dimension $d$, this will increase the computational overhead of our method, as outlined in our reply to reviewer Ng11.
>
> In principle, last-layer retraining can work in combination with (larger) architectures and vision transformers, and even tends to perform better in such cases (see [2]).
>
> >Can the authors include an example with large-scale datasets, such as ImageNet-1K, or a medical imaging benchmark (e.g., ISIC or MedMNIST) to enhance the impact of the proposed algorithms?
>
> We thank the reviewer for raising this suggestion. The datasets used in the manuscript are standard benchmarks for evaluating sub-population shifts (see, e.g., [3]). To the best of our knowledge, a large-scale dataset such as ImageNet-1K is not a standard benchmark in this literature, and it appears not straightforward to evaluate sub-population shifts for this dataset. We are currently searching for a dataset in the medical domain which is relevant for the domain of sub-population shift (e.g. CheXpert [4]), and if time permits will add results for this.
>
> [1] Badr Youbi Idrissi, Martin Arjovsky, Mohammad Pezeshki, and David Lopez-Paz. Simple data balancing achieves competitive worst-group-accuracy. In Bernhard Scholkopf, Caroline Uhler and Kun Zhang (eds.), Proceedings of the First Conference on Causal Learning and Reasoning, volume 177 of Proceedings of Machine Learning Research, pp. 336–351. PMLR, 11–13 Apr 2022.
>
> [2] Pavel Izmailov, Polina Kirichenko, Nate Gruver, and Andrew G Wilson. On feature learning in the presence of spurious correlations. In S. Koyejo, S. Mohamed, A. Agarwal, D. Belgrave, K. Cho, and A. Oh (eds.), Advances in Neural Information Processing Systems, volume 35, pp. 38516–38532. Curran Associates, Inc.,2022
>
> [3] Yuzhe Yang, Haoran Zhang, Dina Katabi, and Marzyeh Ghassemi. Change is hard: a closer look at subpopulation shift. In Proceedings of the 40th International Conference on Machine Learning (ICML'23), Vol. 202. JMLR.org, Article 1652, 39584–39622, 2023
>
> [4] Irvin, J., Rajpurkar, P., Ko, M., Yu, Y., Ciurea-Ilcus, S., Chute, C., Marklund, H., Haghgoo, B., Ball, R., Shpanskaya, K., et al. Chexpert: A large chest radiograph dataset with uncertainty labels and expert comparison. Proceedings of the AAAI conference on artificial intelligence, volume 33, pp. 590–597, 2019

---

> > ### Comment · Reviewer_xwkt · 2024-11-15
> > **Post-Rebuttal**
> >
> > Thank you for addressing my concerns. I believe they have been adequately resolved. As a result, I will be raising my score and recommend acceptance.

---

### Official Review · Reviewer_Ng11 · 2024-11-04

**Soundness:** 3
**Presentation:** 3
**Contribution:** 3
**Rating:** 6
**Confidence:** 2

**Summary:**

The paper addresses the challenge of distribution shifts between training and test data, specifically sub-population shifts. The authors propose a bi-level optimization framework to optimize importance weights while accounting for the finite sample size, thereby improving generalization. The method is validated on benchmark datasets from vision and NLP domains, demonstrating superior performance over traditional importance weighting methods.

**Strengths:**

1. The bi-level optimization of importance weights is novel, addressing the bias-variance trade-off more effectively than heuristic-based weighting.
2. The analysis using a linear regression model provides clear insights into the bias-variance tradeoff, making the proposed method more convincing.
3. The experiment section is comprehensive, covering multiple datasets.
4. The proposed method is useful in practice.

**Weaknesses:**

1. The computational overhead of the proposed methods should be discussed. Since we only need to train on the last layer, I think the computational overhead is not large?
2. The proposed method need to separate the training dataset into val dataset and remaining train dataset. What's the tradeoff here? What's the best way to split? If the training dataset is limited, will the proposed method work?

**Questions:**

see weakness section

---

> ### Author Response · Authors · 2024-11-15
>
> We are grateful to reviewer Ng11 for examining the paper and their feedback. Below, we address the feedback point-by-point.
>
> >The computational overhead of the proposed methods should be discussed.
>
> We thank the reviewer for pointing out the need to discuss the computational overhead of the optimization procedure for the weights. We will first detail the computational resources used for the paper, as well as the approximate time it takes to optimize the weights. We then analyse the algorithmic complexity of the optimization procedure.
>
> For the results in the manuscript, three types of computations were needed:
> 1. Each deep neural network (DNN) is finetuned and, after finetuning, the last-layer embeddings are computed and stored.
> 2. Optimization of the weights for the respective importance weighting procedure.
> 3. Fitting a logistic regression on the last-layer embeddings, using either standard or optimized weights.
> Steps 1 and 3 are also needed when using a standard set of weights for importance weighting. For step 1 we used a single GPU. The memory required for storing the embeddings grows as a function of the total datapoints $n$ and the number of features $d$. For step 3 a single CPU of a Macbook Pro laptop (M2) was used. In our experiments, fitting this logistic regression took less than a minute.
>
> The novelty of our paper is in step 2, for which we also used a single CPU of a Macbook Pro laptop (M2). In our experiments, the additional computational resources required for optimizing the weights were relatively manageable. To give an indication, optimizing the weights for group-weighted ERM for the Waterbirds dataset ($n'=4795$) takes approximately 2 minutes for $T=200$ timesteps. For a bigger dataset such as CelebA, where $n'=162,770$, this took approximately 60 minutes for $T=100$ timesteps. For both datasets, the dimension of the embedding space was $d=2048$. The main contributor towards this extended computational time is the re-fitting of the logistic regression at each step (approximately 0.3 seconds for Waterbirds and 30 seconds for CelebA).
>
> Step 2 is described in Algorithm 1 in the manuscript. Its computational overhead scales linearly with the number of iterations $T$. Per iteration, we re-fit the logistic regression using the weights of the previous step, and we determine the hyper-gradient. The re-fitting of the logistic regression is standard and its algorithmic complexity depends on the optimization procedure, but typically will be increasing in $n'$ and $d.$
>
> In the paper we note that the hyper-gradient factorizes into two gradients (see line 269 of the manuscript). Calculating the first factor has an algorithmic complexity of $O(n_{\mathrm{val}}(d+1) )$, since $\hat{\boldsymbol{\theta}}_{n'}$ consists of $d$ coefficients and an intercept. As detailed in Appendix A, in order to calculate the second factor, we need to calculate the Hessian and invert it, which has an algorithmic complexity of $O(n'(d+1)^2 + (d+1)^3).$ As mentioned in Section 7 of the manuscript, this makes it difficult to extend the current form of the optimization procedure beyond last-layer retraining. It is worth pointing out that when using a logistic regression, we have analytical solutions to both parts of the hyper-gradient, allowing for fast computation (see Appendix A).
>
> We will add a paragraph to the updated manuscript that will detail the computational resources, as well as provide details on the algorithmic complexity of the method.

---

> ### Author Response · Authors · 2024-11-15
>
> >The proposed method needs to separate the training dataset into val dataset and remaining train dataset. What's the tradeoff here?
>
> We agree with the reviewer that there is a tradeoff here. The validation loss is used to optimize the weights. A small validation set size $n_{\mathrm{val}}$ might lead to overfitting of the weights. It should be mentioned here that in the case of a sub-population shift with 4 subgroups (which is the case for all experiments in the manuscript) the weight space is effectively 3-dimensional. This makes the risk of overfitting limited. However, there are contexts with many more subgroups (e.g. concept bottleneck models, as mentioned in Section 7), where the risk of overfitting can become an issue.
> A small $n_{\mathrm{val}}$ can also lead to very small minority subgroups in the validation set. The validation loss might then not be a good estimate of the expected loss, troubling the finding of the optimal weights.
>
> On the other hand, a large validation set size $n_{\mathrm{val}}$, relative to the training set size $n'$, also has disadvantages. The training set is used to optimize the model parameters. A too small $n'$ might lead to overfitting. Since the dimension of the parameter space is typically much larger than the dimension of the weight space, we believe the risk of overfitting the model parameters is more severe. In addition, after optimizing the weights the model gets fitted on the entire dataset of size $n$ (as commonly done after selecting hyperparameters and weights), but the weights are optimized for a training set of size $n'.$ The value of the optimal weights is a function of the training set size (as illustrated in Figure 1). If the difference between $n'$ and $n$ is too big, the weights might not be optimal anymore for the final training of the model on a training set of size $n.$
>
> For our experiments in the paper, we maintained the training/validation split that is standard for each benchmark. Similar to previous work [1] on last-layer retraining, we fitted our models on the training dataset of size $n'$, rather than both the training and validation data. The ratio between training and validation is relatively standard for these datasets (approximately 80/20, 89/11 and 70/30 for Waterbirds, CelebA, and multiNLI, respectively).
>
> We are working on an additional experiment that investigates the role of the ratio between the training and validation set, and how this is impacted by the size of the total dataset. We aim to reproduce Figure 3 of the paper for different ratio's between the training and validation set.
>
> [1] Polina Kirichenko, Pavel Izmailov, and Andrew Gordon Wilson. Last layer re-training is sufficient for robustness to spurious correlations. In The Eleventh International Conference on Learning Representations, ICLR 2023, Kigali, Rwanda, May 1-5, 2023.

---

> ### Comment · Reviewer_Ng11 · 2024-11-25
>
> Thanks for the reply. I will maintain my score.

---

### Author Response · Authors · 2024-11-15
**General reply to all reviewers**

We are deeply grateful to all reviewers for their careful consideration of the paper. We also thank them for acknowledging the relevance, novelty, and potential usefulness of our procedure for optimizing importance weighting.
In order to start the discussion phase, we have posted comments to each individual reviewer with detailed replies to specific questions and remarks.

Based on the initial reviews, we intend to include the results of (at least) two additional experiments in an updated version of the manuscript.
1. An experiment that investigates the role of the ratio between training/validation split, and how this interacts with the size of the dataset. This in order to address the concern raised by reviewer Ng11 about the impact of having a limited training/validation set.
2. To address a question from Reviewer PJmP, we have conducted an additional experiment where we vary learning rates ($\eta$) and momentum values ($\gamma$) to assess whether or not our results are dependent on these hyperparameters of the optimization procedure. We present the results in our reply to reviewer PJmP.

We also thank the reviewers PJmp and Xwkt for their suggestion to add a dataset from the medical domain. We are currently searching for a dataset in the medical domain which is relevant for the domain of sub-population shift (e.g. CheXpert [1]), and if time permits will add results for this.

We look forward to further discussing the paper over the next two weeks.

[1] Irvin, J., Rajpurkar, P., Ko, M., Yu, Y., Ciurea-Ilcus, S., Chute, C., Marklund, H., Haghgoo, B., Ball, R., Shpanskaya, K., et al. Chexpert: A large chest radiograph dataset with uncertainty labels and expert comparison. Proceedings of the AAAI conference on artificial intelligence, volume 33, pp. 590–597, 2019.

---

### Author Response · Authors · 2024-11-25
**Updated manuscript, including new experiments.**

As a follow-up of our initial replies to all reviewers on the 15th of November, we have updated the manuscript accordingly, and uploaded a revised version. The key updates include:
* An additional experiment investigating the effect of the training/validation split ratio (Appendix D).
* An experiment showing how one can select the hyperparameters of the optimization procedure (Appendix F.4).
* Clarifications in writing, including (but not limited to) the computational complexity of the method,  the experimental setup, and a discussion of the required knowledge of the test distribution.

We very much appreciate the detailed and constructive feedback from all reviewers that has helped improve our work. We look forward to further discussions and are happy to address any additional questions or concerns the reviewers may have.

---

### Author Response · Authors · 2024-12-04
**General, concluding remarks by the authors at the end of the discussion phase**

To conclude the discussion phase, we would like to reiterate that we are deeply thankful to all reviewers for their attentive consideration of the paper. We also would like to thank the reviewers for acknowledging the relevance and novelty of our work, as well as highlighting it _'is mathematically well-supported’_, that the _‘experiment section is comprehensive’_ and that _‘the topic clearly has potential for high impact’_.

During the discussion phase, we have tried to address the concerns and questions of the reviewers. We are glad to see that several reviewers have expressed their satisfaction with how we handled their initial reviews. Based on the feedback of the reviewers, we have also updated the manuscript. Here is an overview of the key updates:

·        We have added an experiment to illustrate the effect of the ratio between the training and validation set (addressing the concern of reviewer Ng11).

·        We have added an experiment to illustrate how one can select the additional hyperparameters of our method (addressing the concern of reviewer PJmP).

·        We have clarified our experimental procedure in Section 6 (addressing the concern of reviewer xwkt).

·        We have added discussions about the computational resources used for the experiments and the computational complexity of the method (addressing the concern of reviewer Ng11).

·        We have added a paragraph in the section Conclusion & Discussion that discusses the required knowledge of the test distribution (addressing concerns of reviewers xwkt and DCrk).



In response to the suggestions of reviewers PJmP and xwkt, we considered the possibility of applying our procedure to a medical dataset. However, due to time constraints, we were unable to implement this addition. Furthermore, the manuscript already includes applications to three benchmark datasets, encompassing both vision and natural language processing domains, which we believe sufficiently demonstrate the versatility of our approach for now.

Finally, we would also like to thank the reviewers for their engagement during the discussion period. We believe that as a consequence of their feedback and suggestions, the manuscript has greatly improved.

---

### Meta-Review · Area_Chair_1gjS · 2024-12-23

**Metareview:**

This paper develops a novel way to assign importance weights to data to deal with sub-population shift.  The authors argue that existing methods to deal with distribution shift between training and test data underperform because they don't take into account the sample size of the training data.  The authors propose a bi-level optimization scheme where they jointly optimize the importance weights and model parameters.

The reviewers unanimously recommended accept for the paper (8, 8, 6, 6) albeit with relatively low confidence (3, 2, 2, 2).  They found the paper well written, sound, empirically comprehensive and relevant to the community.  The reviewers noted as weaknesses mostly requests for additional details.  One reviewer noted that the method requires access to the test dataset / distribution to estimate importance weights, which is potentially an issue since this is not always available during training.

Given the unanimous recommendation of the reviewers, and not much cited as weaknesses, there doesn't seem to be any reason to override the reviewers.  Thus the recommendation is to accept.

**Additional Comments On Reviewer Discussion:**

The authors responded to all the reviews.  Only one reviewer really provided a clear weakness i.e. that access to the test data / distribution was problematic.  In particular, they had concerns that the empirical comparison to methods that don't require test data is unfair.  The authors responded to this concern by stating that the application of their method to GDRO didn't require access to the test data.  The reviewer seemed convinced enough to raise their score.

---

### Decision · Program_Chairs · 2025-01-22

Accept (Poster)